# A Closer Look at Personalized Fine-Tuning in Heterogeneous Federated Learning

## Abstract

Federated Learning (FL) enables decentralized, privacy-preserving model training but struggles to balance global generalization and local personalization due to non-identical data distributions across clients. Personalized Fine-Tuning (PFT), a popular post-hoc solution, fine-tunes the final global model locally but often overfits to skewed client distributions or fails under domain shifts. We propose adapting Linear Probing followed by full Fine-Tuning (LP-FT)—a principled centralized strategy for alleviating feature distortion (Kumar et al., 2022)—to the FL setting. Through systematic evaluation across seven datasets and six PFT variants, we demonstrate LP-FT's superiority in balancing personalization and generalization. Our analysis uncovers federated feature distortion, a phenomenon where local fine-tuning destabilizes globally learned features, and theoretically characterizes how LP-FT mitigates this via phased parameter updates. We further establish conditions (e.g., partial feature overlap, covariate-concept shift) under which LP-FT outperforms standard fine-tuning, offering actionable guidelines for deploying robust personalization in FL.

## 1 Introduction

Federated Learning (FL) (McMahan et al., 2017) enables collaborative learning from decentralized data while preserving privacy, typically by training a shared global model, referred to as General FL (GFL). However, variations in client data distributions often limit GFL's effectiveness. Personalized FL (PFL) (Kairouz et al., 2021) addresses this by customizing models to individual clients. *Personalized Fine-Tuning* (PFT) (Wu et al., 2022), a simple and practical strategy in the PFL family, is a widely adopted post-hoc, plug-and-play approach to diverse GFL methods. As shown in Fig. 1(a), PFT fine-tunes the final global model from GFL to personalize it. This simple strategy ensures easy implementation and adaptation across FL scenarios.

Unlike *process-integrated PFL* methods (*e.g.*, those involving server-client coordination that modifies the entire federated training process (Deng et al., 2020; Collins et al., 2021) or local training strategies that require iterative server feedback (Karimireddy et al., 2020; Tamirisa et al., 2024)), PFT eliminates the need for costly global-training-dependent adaptations. Instead, it fine-tunes the final GFL model once post-training, ensuring simplicity, broad compatibility, and deployment robustness without redesigning the GFL framework (see Tab. 1). These characteristics establish PFT as a critical fallback strategy when process-integrated PFL approaches prove infeasible — particularly in scenarios where global training protocols are unmodifiable due to infrastructure lock-in or legacy FL infrastructure, or strict coordination agreement constraints (*e.g.*, healthcare systems bound by long-term service agreements).

However, PFT often causes models to overfit on local data, thereby compromising the generalization of FL. This is particularly concerning in critical real-world applications, such as FL across multiple hospitals for disease diagnosis, where a local model must not only perform well on hospital patient data, but also generalize effectively to diverse patient populations that may be encountered on-site in the future (Xu et al., 2021). Therefore, balancing the optimization of individual client performance (personalization) with strong global performance (generalization across all clients) is crucial (Wu et al., 2022; Huang et al., 2024).

Table 1: Comparisons of Process-Integrated PFL vs. Post-Hoc PFT

| Criterion | Process-Integrated PFL | PFT (Post-Hoc) |
|---|---|---|
| **Global training modification** | **Required** (aggregation changes or iterative local training with server feedback) | **None** (algorithm-agnostic) |
| **Implementation Complexity** | **High** (client-server coordination, custom aggregation/regularization) | **Low** (single fine-tuning step, client autonomy, plug-and-play) |
| **Compatibility with GFL** | **Limited** (framework-specific) | **Broad** (process-agnostic) |

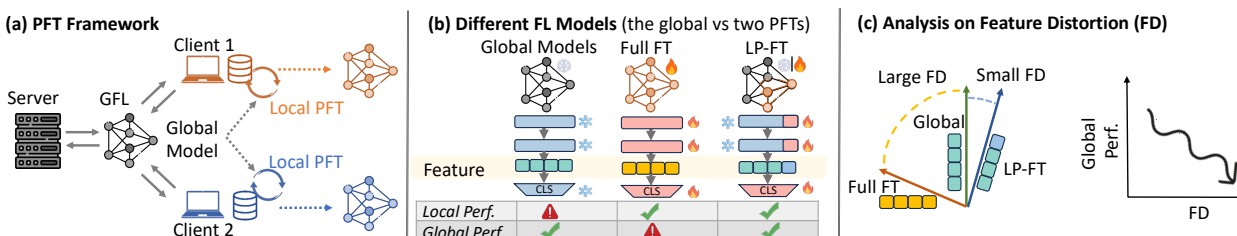

Figure 1: Overview of the problem setting and FL strategies investigated in this paper. (a) PFT framework, where each client fine-tunes a global model trained via GFL (e.g., FedAvg in this paper). Unlike process-integrated PFL, PFT focuses solely on the final fine-tuning stage with no further communication. (b) Three different FL models: the global FL model, the full-parameter FT (full FT) model, and the LP-FT model; their parameter updating patterns and local/global performance (perf.) under data heterogeneity; The fire icon indicates the actively tuned parameter, the frozen icon represents the fixed weight, and the mixed fire-frozen icon denotes the weight that is not actively tuned. (c) Visualization of feature distortion under PFL and its possible link to global generalization.

In this work, we conduct a comprehensive evaluation of various strategies for PFT in heterogeneous FL environments under different distribution shifts, categorized as covariate shift (Peng et al., 2019a; Hendrycks & Dietterich, 2019) and concept shift (Izmailov et al., 2022).

Despite meticulously tuning the hyper-parameters in some FT methods (full parameter FT, sparse FT Lee et al. (2018) and Proximal FT Li et al. (2020b)) adapted in FL, we observe persistent issues of local overfitting when increasing the local fine-tuning epochs, wherein localized performance gains are achieved at the significant cost of global generalization.

LP-FT (Kumar et al., 2022)—a two-phase fine-tuning strategy that first updates *only* the linear classifier (Linear Probing, LP) before optimizing all parameters (Full Fine-Tuning, FT)—has demonstrated state-of-the-art performance in centralized learning by mitigating overfitting and enhancing domain adaptation. However, its potential to address FL challenges, such as client data heterogeneity and instability during decentralized personalization, remains unexplored. In FL, local fine-tuning risks overfitting to client distributions and diverging from globally useful representations. LP-FT's structured separation of updating the head and then fine-tuning offers a principled framework to stabilize personalization in non-IID settings while preserving global knowledge.

Yet, no work has rigorously evaluated LP-FT's efficacy in FL—a critical oversight given the growing demand for lightweight, flexible, and robust personalization strategies. Empirically, we conduct a comprehensive evaluation across seven datasets and diverse distribution shifts, benchmarking our adapted LP-FT against other advanced fine-tuning methods in our PFT framework. Our findings reveal two key insights: (1) existing PFT methods suffer from personalized overfitting, where local fine-tuning distorts feature representations, degrading global performance (Fig. 2); (2) LP-FT mitigates this issue, preserving generalization while enhancing local adaptation under extreme data heterogeneity. Further, extensive ablation studies (Fig. 4) confirm that LP-FT reduces federated feature distortion, establishing it as a strong and scalable baseline for PFT in FL.

Theoretically, we revisit *feature distortion*–a key challenge previously defined in centralized LP-FT as feature shifts under out-of-domain fine-tuning—in FL's unique setting of partially overlapping local and global distributions. Unlike centralized analyses (Kumar et al., 2022), which assume a single ground-truth function, FL involves multiple client-specific ground-truth functions, necessitating a new theoretical framework. We address this by decoupling the feature extractor and classifier to analyze LP-FT's adaptation to heterogeneous client data. Further, we introduce a combined covariate-concept shift setting, better reflecting real-world FL scenarios. Our analysis reveals conditions under which LP-FT outperforms full fine-tuning, advancing the understanding of fine-tuning strategies in FL.

This paper takes a closer look at PFT and establishes LP-FT as a theoretically grounded and empirically viable solution for FL's unique constraints. In summary, our contributions are threefold: (1) Methodologically, this paper presents the first systematic and in-depth study on the post-hoc and plug-and-play PFT framework and introduces LP-FT as an effective approach for handling diverse distribution shifts. We comprehensively demonstrate its ability to balance personalization and generalization in the FL setting. (2) Empirically, we conduct extensive experiments across seven datasets under various distribution shifts, complemented by thorough ablation studies. Our results validate the robustness of LP-FT and reveal overfitting tendencies in prior PFT methods. These empirical insights not only establish LP-FT as a strong baseline for PFT but also provide a foundation for future research in simple and flexible FL personalization. (3) Theoretically, we offer a rigorous theoretical analysis of LP-FT using two-layer linear networks, demonstrating its superior ability to preserve global performance compared to FT in both concept shift and combined concept-covariate shift scenarios.

## 2 Related Work

**Fine-Tuning** pre-trained models has gained prominence in centralized learning, particularly with the rise of foundation models (Bommasani et al., 2021). However, fine-tuning with limited data often leads to overfitting. *Model soups* (Wortsman et al., 2022b) and partial fine-tuning (Lee et al., 2023) further enhance adaptation by selectively updating model components. LP-FT (Kumar et al., 2022), which combines linear probing with full fine-tuning, addresses feature distortions and provides insights into model adaptation under diverse shifts (Trivedi et al., 2023). However, the effectiveness of these centralized fine-tuning strategies in the heterogeneous FL setting remains largely underexplored.

**Personalized FL** aims to address the challenges of decentralized learning with non-IID data. Classical *general FL (GFL)* methods, such as FedAvg (McMahan et al., 2017), struggle in such settings. Despite the advancements in GFL methods (*e.g.*, FedNova (Wang et al., 2020)), FedProx (Li et al., 2020a), Scaffold (Karimireddy et al., 2020)), their focus on building a single global model does not adequately address the data heterogeneity inherent in FL, leading to the emergence of *personalized FL (PFL)* (Ghosh et al., 2020; Yu et al., 2020), which focuses on tailoring individualized models for each client. However, most PFL methods are *process-integrated*, requiring modifications to the global training pipeline through server-client coordination (Deng et al., 2020; Collins et al., 2021) or iterative local training with server feedback (Karimireddy et al., 2020; Tamirisa et al., 2024), or modifying training with customized clustering/regularization (Guo et al., 2024; Son et al., 2024). These approaches impose constraints on flexibility and deployment, as we summarized in Tab. 1. In contrast, *post-hoc personalized fine-tuning (PFT)* Wu et al. (2022) fine-tunes the final global model from GFL without modifying the training process, providing a lightweight and flexible approach for FL personalization. However, its potential is underexplored, possibly due to overfitting risks on client data. Additional discussion on personalization and fine-tuning is in App. B.

## 3 Empirical Study of PFT

To systematically investigate the challenges and opportunities in PFT, we present a comprehensive empirical study. First, in Sec. 3.1, we formalize the problem of PFT and characterize the spectrum of data heterogeneity to be studied. Next, Sec. 3.2 details our experimental setup, including datasets and PFT strategies under consideration. Our investigation then addresses a critical yet understudied phenomenon: Sec. 3.3 analyzes the prevalence of *personalized overfitting* in PFT across distribution shifts, even with careful hyper-parameter

tuning. Motivated by this finding, Sec. 3.4 introduces LP-FT and benchmarks its performance against alternative PFT strategies in FL, showing its superior ability to balance local adaptation with global knowledge retention. Finally, to uncover the mechanistic drivers of generalization challenges, Sec. 3.6 conducts the first systematic analysis of federated feature distortion—quantifying how client-specific fine-tuning trajectories alter latent representations and degrade model robustness.

## 3.1 Overview and Definitions

**Problem Setting.** In a FL setting, each client $i \in [C]$ has a local dataset $(\mathbf{X}_i, \mathbf{Y}_i)$ generated from a potentially distinct distribution, which may differ across clients due to distribution shifts. PFT aims to optimize local model parameters $\theta_L$ for each client, initialized from a well-trained global model $\theta_G$. The objective is to minimize the local loss $\mathcal{L}_L(\theta_L)$ for improved local performance while ensuring that the global loss $\mathcal{L}_G(\theta_L)$ remains close to that of a pre-trained global model. This creates a trade-off between personalization (minimizing local loss) and maintaining generalization (minimizing global loss) across clients. The global data distribution $\mathcal{D}_G$ is defined as a mixture of the local distributions $\mathcal{D}_i$, given by $\mathcal{D}_G = \frac{1}{C} \sum_{i \in [C]} \mathcal{D}_i$.

We formally define distributions of interests, concept shift and covariate shift that directly lead to feature shift in heterogeneous FL context[1], following Li et al. (2021a).

**Covariate Shift** refers to variations in the input feature distribution across clients while keeping the conditional distribution of the output given the input consistent. Formally, for any pair of clients $i$ and $j$ with $i \neq j$, the data-generating process is characterized by:

$$P_i(x) \neq P_j(x), \text{ but } P_i(y \mid x) = P_j(y \mid x) \text{ for all } i \neq j.$$

This means that while clients $i$ and $j$ may have different input distributions $P_i(x)$ and $P_j(x)$, the conditional distribution $P(y \mid x)$ remains consistent across all clients.

**Concept Shift** occurs when the conditional relationship between input features and outputs differs across clients, while the input feature distribution remains unchanged. Formally, for any two clients $i$ and $j$ with $i \neq j$, the data-generating process satisfies:

$$P_i(y \mid x) \neq P_j(y \mid x), \text{ but } P_i(x) = P_j(x) \text{ for all } i \neq j.$$

This implies that although all clients share the same input distribution $P(x)$, the conditional distribution $P_i(y \mid x)$ varies, reflecting different mappings between features and labels across clients.

## 3.2 Empirical Analysis Settings

**Datasets with Covariate Shift.** We include `Digit5`, `DomainNet`, `CIFAR10-C`, and `CIFAR100-C`. `Digit5` and `DomainNet` belong to the *feature-shift* subgroup, where the data features represent different subpopulations within the same classes. For example, `Digit5` contains 10-digit images collected from various sources with different backgrounds, such as black-and-white for MNIST and colorful digits for synthetic datasets. `CIFAR10-C` and `CIFAR100-C` fall under the *input-level shift* category, where 50 types of image corruptions are introduced for evaluation. We simulate 50 clients, each corresponding to a specific corruption type, as detailed in previous works Hendrycks & Dietterich (2018); Mintun et al. (2021); Chen et al. (2021). A detailed explanation of the data splitting and its introduction is provided in Tab. 6 in Appendix. The visualizations of data are provided in Fig. 5.

**Datasets with Concept Shift.** `CheXpert` and `CelebA` are included for this part, whereas both belong to the *spurious correlation-based shift* subgroup, which involves misleading relationships in the training data that models may exploit, despite being unrelated to the actual target. This reliance can lead to poor performance when such correlations are absent in new data, classifying it as a form of concept shift (Izmailov et al., 2022). Similarly, Tab. 6 and Fig. 5 provide further details.

**Fine-tuning Strategies.** Our study focuses on post-hoc PFT, a plug-and-play framework that operates exclusively after GFL training. Unlike conventional fine-tuning in centralized settings that primarily addresses

---

[1]We also realized that LP-FT can be effective for label shift settings as the results shown in App. 3.5.

domain adaptation by transferring a model from a source to a disjoint target domain, PFT operates on a global model pre-trained via GFL, which has already been exposed to heterogeneous client data during collaborative training and must balance local performance (adapting to a client's unique distribution) with global performance (avoiding overfitting to statistically biased local updates and preserving cross-client generalizability).

In this study, we establish a suite of fine-tuning strategies that can be easily integrated into PFL as **baselines** for PFT: *Full-parameter FT* is a naive FT strategy. It adjusts all model parameters. *Proximal FT* (Li et al., 2020b) aims to preserve the pre-trained model's original knowledge. It applies proximal regularization to penalize large deviations from the initial model parameters, helping to maintain generalization. *Sparse FT* (Lee et al., 2018) promotes sparsity in parameter updates. It adjusts only the most relevant weights, enhancing efficiency while regularizing the training from overfitting. *Soup FT* (Wortsman et al., 2022a) improves robustness by averaging the weights of multiple fine-tuned model instances. Each instance is trained with different initializations, creating a "model soup" that integrates their strengths. *LSS FT* (Chen et al., 2024) (Local Superior Soups) is an innovative model interpolation-based local training technique designed to enhance FL generalization and communication efficiency by encouraging the exploration of a connected low-loss basin through optimizable and regularized model interpolation. Each strategy is designed to balance model performance with different priorities, such as preserving knowledge, enhancing robustness, or improving efficiency. A more detailed experiment setting is presented in App. C.[2]

### 3.3 Global and Local Performance Trends in PFT Baselines

In practice, PFT is susceptible to overfitting to local data, due to the relatively small amount of data available at local clients. Note that the *overfitting* defined in the FL context is characterized by *a consistent improvement in local performance while global performance noticeably deteriorates (Wu et al., 2022; Chen et al., 2023) – the average gain in local performance can be smaller than the loss in global performance.* To measure the model's overall local and global performance, we measure the averaged client-wise local and global accuracy. Specifically, this metric reflects the average performance between clients' *local* test accuracy and their local model's accuracy on the rest of the clients (*global* accuracy). The metric's decreasing trend with increasing local training epochs during the finetuning stage indicates personalized overfitting. Notably, this trend persists even when considering only global performance metrics, as local performance tends to show increases in PFT under overfitting conditions.

To make these notions precise, let $D_i^{\text{test}}$ denote the test dataset associated with client $i$, and let $f_{v_i, B_i}$ denote the personalized model obtained for client $i$ after applying post-hoc PFT. The *local accuracy* is defined as the classification accuracy of the personalized model $f_{v_i, B_i}$ when evaluated on the client's own test data $D_i^{\text{test}}$, namely,

$$\text{Local} = \frac{1}{C} \sum_{i=1}^{C} \frac{1}{|D_i^{\text{test}}|} \sum_{(x,y) \in D_i^{\text{test}}} \mathbf{1}\{f_{v_i, B_i}(x) = y\},$$

and the *global accuracy* is defined as

$$\text{Global} = \frac{1}{C(C-1)} \sum_{i=1}^{C} \sum_{j \neq i} \frac{1}{|D_j^{\text{test}}|} \sum_{(x,y) \in D_j^{\text{test}}} \mathbf{1}\{f_{v_i, B_i}(x) = y\}.$$

In all subplots of Fig. 2, we evaluate baseline PFT strategies under diverse distribution shifts, including input-level shifts (`CIFAR100-C`), feature-level shifts (`Digit5`), and spurious correlation-based shifts (`CheXpert`). We systematically adjusted hyperparameters to evaluate their impact on performance. Fig. 2a demonstrates that overfitting persists even when fine-tuning with reduced learning rates. Fig. 2b reveals that gradient sparsity adjustments (where higher sparsity rates mask more parameter updates) fail to mitigate overfitting as training epochs increase. Fig. 2c further shows that proximal regularization terms, designed to bias updates toward the initial global model, still exhibit global performance decay despite regularization.

---

[2]We primarily focus on CNN-based models. We also include parameter-efficient fine-tuning results on transformer-based models in Appendix Tab. 7.

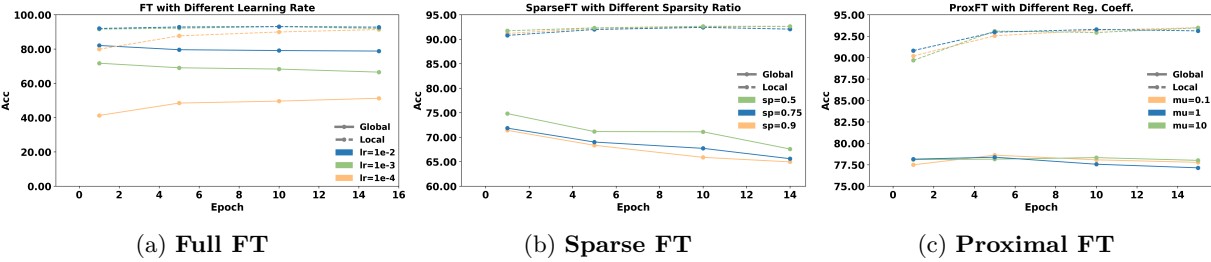

(a) **Full FT**        (b) **Sparse FT**        (c) **Proximal FT**

Figure 2: Visualization of the prevalence of personalization overfitting across different distribution shift scenarios, where (a) shows the global and local accuracy under different learning rates for full-parameter fine-tune; (b) shows the different sparsity rate for sparse fine-tune; (c) shows the different regularization strength under the proximal fine-tune. In all subfigures, the global accuracy is shown as the solid line, and the local accuracy is shown as the dashed line. As shown, global accuracy consistently declines while local accuracy either increases or remains stable across different hyperparameter settings. This suggests that PFT baseline methods are prone to overfitting, even with careful hyperparameter tuning.

## 3.4 Performance Comparison

**Linear Probing then Fine-Tuning.** To address the challenge of personalized overfitting in conventional fine-tuning methods within PFT, we propose a simple yet effective approach through Linear Probing followed by Fine-Tuning (LP-FT) for FL. The idea is motivated by LP-FT (Kumar et al., 2022)—a two-phase fine-tuning strategy in centralized training that first updates *only* the linear classifier (Linear Probing, LP) before optimizing all parameters (Full Fine-Tuning, FT) to improve out-of-domain performance while preserving in-domain performance. We adapt the strategy in PFT as follows: *In practice, clients initialize weights from the model after GFL, first perform linear probing, and then fine-tune the full model as shown in Fig. 1 (b).* This LP-FT approach achieves strong personalization while maintaining generalizability across diverse clients.

**Experimental Settings.** To isolate the impact of PFT strategies and avoid conflating gains from GFL optimization, we standardize the GFL stage by fixing its method to FedAvg, the foundational and most widely used GFL method. Within this framework, we focus on comparing different *post-hoc* FT methods to demonstrate the effectiveness of LP-FT in PFT (see Fig. 1 (a)). After the GFL stage, all the clients further fine-tune the obtained global model using local data for 15 epochs for personalization. The final models are evaluated using the *metrics* described below. Details of the datasets, preprocessing steps, data splitting, and models used are provided in App. C.3, Tab. 6.

**Metrics.** We adapt five metrics in our baseline experiments: *(1) Local Accuracy (Local)* measures the performance of the PFT model on the client's local test set. Higher *Local Acc* indicates better personalization. *(2) Global Accuracy (Global)* measures the PFT model's average test accuracy over all other clients' test sets. Higher *Global Acc* indicates better generalization. *(3) Client-wise Standard Deviation (C-Std.)* calculates the standard deviation of local test accuracies across all clients. Lower *C-Std.* indicates less variance in performance among clients. *(4) Worst Accuracy (Worst)* reports the lowest test accuracy among all clients. The closer this value is to *Local Acc*, the better the worst-case generalization. *(5) Average* reports the average of both *Local Acc* and *Global Acc*, providing a better understanding of the tradeoff between personalization (local performance) and generalization (global performance). All metrics, except *C-Std.*, are averaged over the number of clients, and higher values are preferable. For *C-Std.*, lower values are better.

**Results.** Our results are presented in Tab. 2, where the best method is highlighted in **bold**. Datasets with the same distribution shift pattern are grouped into the same colors as detailed in the caption. Tab. 2 shows that LP-FT consistently achieves the highest global and average accuracy across most datasets, demonstrating strong generalization and personalization performances, particularly in challenging conditions like `CIFAR100-C` and `CIFAR10-C`. Sparse FT also performs well, especially in `Digits5` and `DomainNet`, but generally lags behind LP-FT. LSS FT, Soup FT and Proximal FT show mixed results, with stronger performance in specific datasets such as `CheXpert` but weaker overall compared to LP-FT. Standard fine-tuning consistently underperforms, highlighting the limitations of basic fine-tuning methods in heterogeneous data scenarios.

Table 2: Performance of various PFT strategies. **Red** represents the *input shift* subgroup; **green** from the *feature-shift* subgroup; **blue** the *spurious correlation-based shift* subgroup. Each experiment is performed three times independently with different random seeds, and the standard deviation of the results is presented in parentheses. ↑ indicates that higher values are better, while ↓ indicates that lower values are better.

| Dataset | Method | Local ↑ | Global ↑ | C-Std. ↓ | Worst ↑ | Average ↑ |
|---|---|---|---|---|---|---|
| **CIFAR10-C** | FT | 54.50 (0.64) | 44.16 (0.13) | **10.04 (0.06)** | 19.83 (0.18) | 39.50 (0.33) |
| | Proximal FT | 61.76 (0.13) | 53.58 (0.14) | 11.61 (0.08) | 25.82 (0.12) | 47.05 (0.07) |
| | Soup FT | 56.36 (0.23) | 44.94 (0.06) | 10.22 (0.06) | 20.47 (0.35) | 40.59 (0.09) |
| | Sparse FT | 61.31 (0.01) | 50.21 (0.17) | 11.10 (0.11) | 24.56 (0.09) | 45.36 (0.04) |
| | LSS FT | 56.21 (0.33) | 46.81 (0.04) | 10.05 (0.08) | 21.61 (0.37) | 43.67 (0.08) |
| | LP-FT | **63.55 (0.04)** | **55.35 (0.01)** | 12.45 (0.01) | **26.33 (0.06)** | **48.41 (0.03)** |
| **CIFAR100-C** | FT | 20.05 (0.05) | 14.45 (0.04) | **5.37 (0.02)** | 3.37 (0.06) | 12.62 (0.03) |
| | Proximal FT | 27.38 (0.15) | 19.96 (0.11) | 6.90 (0.04) | 4.84 (0.04) | 17.41 (0.05) |
| | Soup FT | 20.99 (0.24) | 14.81 (0.04) | 5.48 (0.03) | 3.56 (0.01) | 13.12 (0.06) |
| | Sparse FT | 28.93 (0.04) | 20.66 (0.02) | 7.75 (0.02) | 5.05 (0.09) | 18.15 (0.10) |
| | LSS FT | 20.54 (0.19) | 15.42 (0.03) | 5.32 (0.03) | 3.62 (0.01) | 14.22 (0.06) |
| | LP-FT | **32.60 (0.14)** | **25.44 (0.10)** | 9.66 (0.04) | **5.92 (0.06)** | **21.32 (0.04)** |
| **Digit5** | FT | 91.17 (0.90) | 67.87 (0.74) | 22.93 (0.28) | 42.03 (0.48) | 67.02 (0.70) |
| | Proximal FT | **92.09 (0.18)** | 81.40 (0.03) | 15.04 (0.15) | 61.71 (0.16) | 78.40 (0.09) |
| | Soup FT | 91.82 (0.34) | 70.82 (0.43) | 21.99 (0.67) | 45.10 (1.27) | 69.02 (0.65) |
| | Sparse FT | 91.43 (0.31) | 76.89 (0.72) | 17.90 (0.38) | 54.21 (0.56) | 74.21 (0.35) |
| | LSS FT | 91.59 (0.28) | 73.13 (0.30) | 22.04 (0.53) | 45.32 (1.13) | 71.15 (0.53) |
| | LP-FT | 91.20 (0.04) | **82.78 (0.05)** | **13.75 (0.02)** | **65.80 (0.02)** | **79.92 (0.02)** |
| **DomainNet** | FT | 64.90 (1.18) | 42.48 (0.58) | 17.49 (0.75) | 22.31 (0.93) | 43.23 (0.52) |
| | Proximal FT | 67.20 (1.39) | 56.05 (0.27) | **16.68 (0.36)** | 33.20 (1.79) | 52.60 (0.35) |
| | Soup FT | 67.48 (0.61) | 44.27 (0.46) | 18.44 (0.42) | 23.73 (1.24) | 44.49 (0.54) |
| | Sparse FT | **69.62 (0.53)** | 50.24 (0.44) | 18.14 (0.17) | 27.89 (0.15) | 49.14 (0.45) |
| | LSS FT | 66.37 (0.53) | 45.34 (0.40) | 18.02 (0.38) | 22.63 (1.05) | 45.75 (0.42) |
| | LP-FT | 68.50 (0.19) | **57.52 (0.20)** | 17.36 (0.21) | **34.53 (0.44)** | **53.52 (0.19)** |
| **CheXpert** | FT | 76.18 (0.41) | 76.25 (0.56) | 0.35 (0.13) | 76.31 (0.76) | 76.25 (0.44) |
| | Proximal FT | 76.44 (0.07) | 76.63 (0.09) | 0.71 (0.09) | 76.81 (0.07) | 76.63 (0.07) |
| | Soup FT | 77.51 (0.15) | 77.49 (0.31) | 0.48 (0.07) | 77.46 (0.43) | 77.49 (0.26) |
| | Sparse FT | 77.29 (0.13) | 77.20 (0.14) | **0.31 (0.11)** | 77.11 (0.25) | 77.20 (0.14) |
| | LSS FT | 77.49 (0.14) | 77.51 (0.28) | 0.52 (0.08) | **77.53 (0.37)** | 77.52 (0.24) |
| | LP-FT | **77.64 (0.37)** | **77.54 (0.37)** | 0.53 (0.41) | 77.43 (0.71) | **77.54 (0.37)** |
| **CelebA** | FT | 90.55 (1.20) | 73.76 (2.15) | 18.79 (3.64) | 53.52 (5.51) | 72.39 (2.84) |
| | Proximal FT | **93.74 (0.59)** | 81.11 (0.82) | 13.39 (1.14) | 67.50 (2.10) | 80.78 (0.90) |
| | Soup FT | 89.42 (2.16) | 75.28 (1.11) | 16.29 (1.19) | 57.79 (2.90) | 74.17 (1.50) |
| | Sparse FT | 91.43 (0.48) | 77.32 (1.46) | 14.16 (2.57) | 62.94 (4.34) | 77.65 (1.65) |
| | LSS FT | 89.17 (2.05) | 77.35 (1.03) | 16.23 (1.28) | 59.64 (2.86) | 76.74 (1.46) |
| | LP-FT | 93.24 (0.17) | **83.32 (0.31)** | **11.18 (0.14)** | **71.89 (0.75)** | **82.82 (0.64)** |

We further compare against (i) PEFT methods (LoRA and Adapters; Appendix Table 7) to contextualize LP-FT beyond the post-hoc PFT baselines above, and (ii) process-integrated PFL methods (Appendix Table 8). We report the full results in the Appendix for space, and to avoid shifting the paper's main focus away from our comprehensive analysis of post-hoc personalization. For (i), we evaluate LoRA- and Adapter-based PEFT with a vision transformer; as shown in Table 7, LP-FT still achieves stronger local and global performance than these PEFT baselines under a transformer architecture. For (ii), we additionally benchmark against methods that modify the global training procedure, while our approach is built on vanilla FedAvg; Table 8 shows that FedAvg + LP-FT outperforms multiple process-integrated PFL pipelines, achieving the best local, global, and average accuracy despite not altering the global training process. Taken together, these results highlight that LP-FT is a simple yet effective, orthogonal personalization step that delivers strong local and global performance compared to both standard global training and PEFT baselines.

### 3.5 Label Shift

In this section, we examine the impact of LP-FT on the label shift scenario. We simulated label shift using a Dirichlet distribution with an alpha parameter set to 0.1. The dataset was distributed across 20 clients, and we trained a ResNet18 model for classification. Results is shown in Table 3.

|  | Baseline | Local | Global | C-Std. | Worst | Avg |
|---|---|---|---|---|---|---|
| **CIFAR10** | FT | 87.20 (0.24) | 16.67 (0.14) | **26.81 (1.47)** | 0.01 (0.00) | 34.62 (0.09) |
|  | Proximal FT | 89.80 (1.21) | 17.42 (1.46) | 27.31 (0.08) | 0.01 (0.00) | 35.75 (0.09) |
|  | Soup FT | 88.16 (0.15) | 16.94 (0.05) | 27.07 (0.11) | 0.01 (0.00) | 35.04 (0.04) |
|  | Sparse FT | 89.16 (0.00) | 17.54 (0.14) | 27.27 (0.02) | 0.01 (0.00) | 35.57 (0.04) |
|  | LP-FT | **90.15 (0.43)** | **17.73 (0.16)** | 27.37 (0.09) | 0.01 (0.00) | **35.96 (0.10)** |

Table 3: Comparison of PFT methods on CIFAR10 label shift setting.

Tab. 3 compares different fine-tuning methods on CIFAR10 across multiple evaluation metrics, including local performance, global performance, robustness to corruption (C-Std.), worst-case performance, and average performance. Among the methods, LP-FT achieves the best results, excelling in local performance (90.15), global evaluation (17.73), and average performance (35.96). Proximal FT and Sparse FT also perform competitively, with improvements over standard fine-tuning (FT) and Soup FT in most metrics. All methods show near-identical performance in the worst-case scenario (0.01), indicating a shared limitation in extreme cases. Overall, LP-FT demonstrates the most robust and effective fine-tuning approach on CIFAR10.

### 3.6 Insight and Explanation on the Observations

Given the unique design of LP-FT, we hypothesize that its superior performance in PFT stems from its ability to mitigate federated feature distortion — a phenomenon where client-specific fine-tuning disrupts the global model's learned representations. We empirically validate this hypothesis through a systematic analysis of feature space dynamics across diverse data heterogeneity scenarios.

**Federated Feature Distortion.** Consider a feature extraction function $f : \mathcal{X} \to \mathbb{R}^k$, which maps inputs from the input space $\mathcal{X}$ to a representation space $\mathbb{R}^k$. Let $\theta_G$ denote the global pre-trained model and $\theta_i$ the fine-tuned model after local fine-tuning for client $i$. Assume there are $C$ clients in total, each with $n$ samples. Let $x_{c,j}$ represent the $j$-th data point of the $c$-th client. The *federated feature distortion* $\Delta_c(f)$ quantifies the change in features after fine-tuning for the $c$-th client, defined as the average $\ell_2$ distance between the representations produced by the global model and the locally fine-tuned model over all data points across all clients. Formally, it is expressed as: $\Delta_c(f) = \frac{1}{n} \sum_{j=1}^{n} \|f(\theta_G; x_{c,j}) - f(\theta_c; x_{c,j})\|_2$, where $\| \cdot \|_2$ is the $\ell_2$ distance in the representation space $\mathbb{R}^k$. We compute the average of $\Delta_c(f)$ across all clients to represent the feature distortion in the PFT setting, as shown in Fig. 3.

**Empirical Validation.** To quantify federated feature distortion, we measure the $\ell_2$ distance between global and locally fine-tuned feature representations using `DomainNet` and `Digit5`. As shown in Fig. 3(a), the full FT method induces severe feature distortion, correlating with a significant drop in global accuracy, whereas LP-FT maintains stable global performance with lower distortion.

To further isolate the effect of feature distortion from local loss magnitude, we apply loss flooding (Ishida et al., 2020) to control local training loss levels (0.1, 0.5, 1.0). Fig. 3(b) shows that at fixed local loss levels, LP-FT consistently outperforms FT in global accuracy, confirming that its advantage stems from reduced feature distortion rather than differences in local optimization dynamics.

## 4 Theoretical Analysis of the LP-FT in FL

Building on our empirical observations in Sec. 3, where LP-FT consistently outperforms baseline PFT methods and demonstrates a significant reduction in federated feature distortion, we now present a theoretical analysis to uncover the mechanistic principles underlying its success. To understand how feature learning impacts

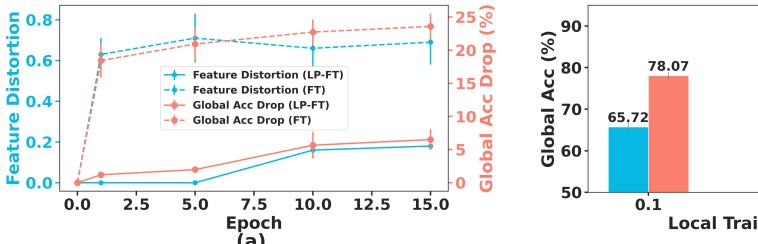

Figure 3: Observations of the feature distortion in PFT setting, where (a) presents the positive correlation between global performance drops and feature distortion intensity on `DomainNet` and (b) presents the ablation study on preserving federated features with controlled local train loss on `Digit5`. We set local loss thresholds (0.1, 0.5, and 1.0) and used gradient ascent when the loss fell below, ensuring training loss fluctuated around these points.

generalization error in PFT, we decompose the data-generating function and the model into two components: a feature extractor and a linear head. This decomposition allows us to distinguish between the learned features and their influence on performance. Specifically, in Sec. 4.1 and Sec. 4.2, we formalize concept and covariate shifts within a two-layer linear network and examine how LP-FT effectively adapts to these shifts, outperforming full-parameter fine-tuning (FT) in FL.

**Overview of Theoretical Analysis:** To compare the performance of LP-FT and FT, we make assumptions about the data-generating function for clients (Assumption 4.1) and a specific model structure (Assumption 4.2). Based on these assumptions, we analyze the global performance of LP-FT and FT under concept shift (Theorem 4.4) and combined concept-covariate shift (Theorem 4.5).

## 4.1 LP-FT's Global Performance Under Concept Shift

In this section, we analyze LP-FT's performance compared to FT under concept shift. To facilitate a rigorous theoretical study, we define the data-generating process and model structure across clients, assuming both are represented by two-layer linear networks, as in (Kumar et al., 2022).

**Assumption 4.1** (Data-Generating Process). The data-generating function for client $i$ is given by $y_i = V_i^{*T} B_* x_i$ for all $i \in [C]$, where $y_i \in \mathbb{R}$, $C$ is the number of clients, $x_i \in \mathbb{R}^d$, $B_* \in \mathbb{R}^{k \times d}$, and $V_i^* \in \mathbb{R}^k$. All clients share a common feature extractor $B_*$, assumed to have orthonormal rows, while their linear heads $V_i^*$ differ. Each $V_i^*$ decomposes as $V_i^* = \begin{bmatrix} V_{com}^{*T} & \lambda e_i^T \end{bmatrix}^T$, where $V_{com}^* \in \mathbb{R}^m$ is shared across clients, $e_i \in \mathbb{R}^C$ is a unit vector, and $\lambda$ controls heterogeneity. Here, $m + C = k$.

This assumption distinguishes between a shared and client-specific component in the data-generating functions, allowing analysis of both global and local performance of PFT methods after fine-tuning.

**Assumption 4.2** (Model Structure). The training model is a two-layer linear network defined as $y = V^T B x$, where $V \in \mathbb{R}^k$ is the linear head and $B \in \mathbb{R}^{k \times d}$ is the feature extractor. The dimensions of $V$ and $B$ match Assumption 4.1, allowing the model to learn both shared and client-specific data components.

While real-world federated learning systems typically employ deep nonlinear models, it is standard in theoretical analyses to study simplified architectures—such as two-layer or deep linear networks—to obtain tractable insights under data heterogeneity (Charles & Konečnỳ, 2021; Huang et al., 2021; Collins et al., 2022). Under Assumptions 4.1 and 4.2, this abstraction allows us to isolate the interaction between shared representations and client-specific heads and to precisely characterize how local updates affect global performance. Our objective is not to provide architecture-agnostic guarantees, but rather to identify the core mechanisms driving federated feature distortion within this setting. Accordingly, the analysis that follows offers a principled explanation for LP-FT's observed advantage.

In PFT settings, our objective is to evaluate the performance of a model on both global and local data. By local data, we refer to the data of a specific client undergoing fine-tuning (e.g., client $i$). The local and global

losses are defined using the Mean Squared Error (MSE) as follows:

$$\mathcal{L}_L(V, B) = \mathbb{E}_{(x,y)\sim\mathcal{D}_i}\big[\frac{1}{2}(V^T Bx - y)^2\big] = \mathbb{E}_{x\sim\mathcal{D}_i}\big[\frac{1}{2}\big(V^T Bx - V_i^{*T}B_* x\big)^2\big],$$

$$\mathcal{L}_G(V, B) = \mathbb{E}_{(x,y)\sim\mathcal{D}_G}\big[\frac{1}{2}(V^T Bx - y)^2\big] = \frac{1}{C}\sum_{i\in[C]}\mathbb{E}_{x\sim\mathcal{D}_i}\big[\frac{1}{2}\big(V^T Bx - V_i^{*T}B_* x\big)^2\big].$$

Since this section focuses on concept shift, we assume all clients' data is drawn from similar distributions. Accordingly, we assume for every client $i \in [C]$, the input features satisfy $\mathbb{E}_{x\sim\mathcal{D}_i}[xx^T] = I_d$. This isotropy assumption facilitates the analysis of high-dimensional phenomena—such as optimization dynamics and generalization behavior—and is commonly adopted in related theoretical work (Ghorbani et al., 2019; Hastie et al., 2022).

With the theoretical framework established by Assumptions 4.1 and 4.2, we compare the global performance of LP-FT and FT, highlighting cases where LP-FT outperforms FT. As demonstrated in Collins et al. (2022) FedAvg learns a shared data representation among clients if such a common representation exists. In a PFT setting, the initial model is trained on data from all clients to capture their shared components. Thus, we initialize the model parameters as $B_0 = B_*$ and $V_0 = \begin{bmatrix} V_{com}^{*}{}^T & \mathbf{0} \end{bmatrix}^T$. The assumption $B_0 = B_*$ is consistent with Proposition 3.7 of Kumar et al. (2022), where a similar initialization is adopted to compare LP-FT and FT. In LP-FT, a step of linear probing first updates $V_0$ using local data while keeping $B_0$ fixed, followed by full fine-tuning to update both $V$ and $B$. In contrast, FT performs only the second step. The following lemma characterizes $B$ after one gradient descent step in FT, forming the basis for our comparison.

**Lemma 4.3.** *Under Assumptions 4.1 and 4.2, and assuming that $\mathbb{E}_{x\sim\mathcal{D}_i}[xx^T] = I_d$ for all clients $i \in [C]$, let the initial parameters before starting FT be $B_0 = B_*$ and $V_0 = \begin{bmatrix} V_{com}^{*}{}^T & \mathbf{0} \end{bmatrix}^T$. Assume fine-tuning is performed locally on the data of the $i$-th client. Let $B_{FT}$ denote the feature extractor matrix after a single gradient descent step (processing the entire dataset once) with learning rate $\eta$. If $(b_j^{FT})^T$ is the $j$-th row of $B_{FT}$, then:*

$$\mathbb{E}\Big[(b_j^{FT})^T\Big] = (b_j^*)^T + \eta\lambda(V_0)_j(b_{m+i}^*)^T,$$

*where $(b_j^*)^T$ is the $j$-th row of $B_*$ , and $(V_0)_j$ is the $j$-th element of $V_0$ for $j \in [k]$.*

This lemma examines the impact of FT on the feature extractor $B_{FT}$, highlighting the deviations from the pre-trained matrix $B_0 = B_*$. Given that all clients share the same $B_*$ in their labeling functions, substantial changes to the feature extractor can degrade global performance. Since the matrix $B$ functions as the feature extractor in our framework, significant feature distortion occurs when $B_{FT}$ deviates considerably from $B_*$. Building on Lemma 4.3, Theorem 4.4 offers a comparative analysis of the global performance of LP-FT versus FT in the context of concept shift.

**Theorem 4.4.** *Under Assumptions 4.1 and 4.2, and assuming $\mathbb{E}_{x\sim\mathcal{D}_i}[xx^T] = I_d$ for all clients $i \in [C]$, let the initial model parameters be $B_0 = B_*$ and $V_0 = \begin{bmatrix} V_{com}^{*}{}^T & \mathbf{0} \end{bmatrix}^T$. Let $B_{FT}$ and $V_{FT}$ denote the parameters of the FT method after one gradient descent step (processing the entire dataset once). For LP-FT, let $B_{LPFT}$ and $V_{LPFT}$ denote the parameters after (i) linear probing, which optimizes $V$ with $B$ fixed at $B_*$, and (ii) one gradient descent step with learning rate $\eta$. Then:*

$$\mathcal{L}_G(V_{LPFT}, B_{LPFT}) \leq \mathcal{L}_G(V_{FT}, B_{FT}).$$

This theorem characterizes the global performance of LP-FT, suggesting that under concept shift, LP-FT achieves better performance on global data than FT. When starting from a model initialized to capture the shared feature extractor and linear head among clients, LP-FT is more effective in minimizing global loss, aligning with common FL scenarios where the initial model leverages shared client structure.

## 4.2 LP-FT's Global Performance under Combined Concept and Covariate Shifts

In the previous section, we assumed all clients' data came from the same distribution with $\mathbb{E}_{x\sim\mathcal{D}_i}[xx^T] = I_d$. However, this may not hold in many practical scenarios. To address this, we introduce covariate shift, where

each client's data is generated as $x_i = e_i + \epsilon z$, with $z \sim \mathcal{N}(0, I)$, $e_i$ as a client-specific shift, and $\epsilon$ controlling the noise level. This extension captures the non-iid nature of data among clients and provides a framework to model data heterogeneity. The model structure and data-generating assumptions remain consistent with Sec. 4.1. This section thus considers both concept and covariate shifts. Theorem 4.5 analyzes the impact of heterogeneity on the global performance of LP-FT and FT.

**Theorem 4.5.** *Under Assumptions 4.1 and 4.2, let each client's data be $x_i = e_i + \epsilon z$, where $z \sim \mathcal{N}(0, I)$ and $e_i$ is a client-specific shift. Assume the initial parameters are $B_0 = B_*$ and $V_0 = \begin{bmatrix} V_{com}^{*T} & \mathbf{0} \end{bmatrix}^T$. Let $B_{FT}, V_{FT}$ be the FT parameters after one gradient descent step, and $B_{LPFT}, V_{LPFT}$ be the LP-FT parameters after linear probing and one gradient descent step (with learning rate $\eta$). Then, there exists a threshold $\lambda^*$ such that for all $\lambda \leq \lambda^*$:*

$$\mathcal{L}_G(V_{LPFT}, B_{LPFT}) \leq \mathcal{L}_G(V_{FT}, B_{FT}).$$

*Remark* 4.6. In Theorem 4.5, the parameter $\lambda$ characterizes the level of heterogeneity among clients. The theorem shows that under both covariate and concept shifts, LP-FT outperforms FT in low heterogeneity settings ($\lambda \leq \lambda^*$), highlighting its advantage in maintaining generalization. To further reinforce the theoretical insights and cover more extensive settings, App. 5 provides extensive empirical validation, confirming the global superiority of LP-FT over FT under combined concept-covariate shifts. While the theoretical analysis in Theorem 4.5 focuses on the low heterogeneity regime, the experiments in App. 5 explore a broader range, including both high and low heterogeneity levels. Notably, LP-FT consistently outperforms FT across all heterogeneity regimes, aligning with our theoretical results in Sec. 4.2, particularly for deep neural networks in realistic PFT settings. These findings validate and extend our theoretical insights, demonstrating LP-FT's robustness and superiority in diverse distribution shift scenarios (see also App. F).

*Remark* 4.7. Equivalent results of Theorems 4.4 and 4.5 hold at the local level (see Corollaries E.1 and E.2). This follows from the fact that the global loss is defined as the average of client-specific local losses, so fixing a client and isolating its contribution preserves the same ordering.

# 5 Further Empirical Validations for Theoretical Findings

Table 4: Performance under label-flipping for FT and LP-FT across different label-flipping ratios (LF.R.).

| Metric | LF.R. 20% | LF.R. 30% | LF.R. 40% | LF.R. 50% |
|---|---|---|---|---|
| **FT Avg.** $\uparrow$ | 67.73 | 60.04 | 58.27 | 60.06 |
| **LP-FT Avg.** $\uparrow$ | 79.83 | 72.95 | 71.55 | 73.26 |
| **FT Global** $\uparrow$ | 68.76 | 55.18 | 53.70 | 56.17 |
| **LP-FT Global** $\uparrow$ | 83.08 | 72.75 | 69.89 | 72.32 |
| **FT Local** $\uparrow$ | 91.32 | 91.12 | 90.84 | 91.88 |
| **LP-FT Local** $\uparrow$ | 91.23 | 89.20 | 90.02 | 90.87 |

Despite being based on simplified data and model assumptions, our theoretical results demonstrate significant practical relevance. In this section, we empirically validate the contributions in Sec. 4, exploring the performance implications of controllable heterogeneities in neural networks and datasets.

**Experimental Settings.** To validate the impact of $\lambda$ in Theorem 4.5, we simulate a controllable concept shift setting on the `Digit5` dataset with label-flipping under PFT for both FT and LP-FT. For each client, a proportion of labels is randomly flipped, referred to as the flipping ratio. For example, class one is flipped with class two for the first client, and class two with class three for the second, using a randomized mechanism. A higher flipping ratio indicates greater heterogeneity $\lambda$. The settings align with prior studies: the model is pre-trained within the FL framework and used to initialize both FT and LP-FT. This simulates the combined concept-covariate shift discussed in Sec. 4.2. Flipping labels reflects different labeling functions, where higher flipping rates indicate stronger concept shifts. The `Digit5` dataset also introduces covariate shift, as outlined in Sec. 3.2.

**Results.** As shown in Tab. 4, LP-FT consistently outperforms FT in global performance across various flipping ratios. This aligns with our theoretical results in Sec. 4.2, especially for deep neural networks under realistic PFT settings. The flipping rate controls concept shift heterogeneity, with higher rates indicating greater heterogeneity, while varying data distributions introduce covariate shift. These experiments simulate the combined concept-covariate shift, as analyzed in our framework. Notably, LP-FT outperforms FT in all heterogeneity levels, validating its advantage in both low and high heterogeneity regimes (larger flipping ratios).

## 6 Conclusion

In this work, we studied an important PFL paradigm – PFT and tackled its key challenge of balancing local personalization and global generalization. We establish LP-FT as a theoretically grounded and empirically robust solution for PFT. Our work demonstrates that LP-FT effectively mitigates federated feature distortion, balancing client-specific adaptation with global generalization under extreme data heterogeneity. Methodologically, we are the first to adapt LP-FT to post-hoc PFT; empirically, we validate LP-FT's superiority across seven datasets; theoretically, we formalize its advantages in FL's unique covariate-concept shift regime. This work advances lightweight, deployable personalization for real-world FL systems.

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

# Contents

**Roadmap of Appendix.** This appendix provides all supplemental material referred to in the main paper. First, in Section A we review related work on personalized federated learning (FL) and fine-tuning. Section B introduces the variants of fine-tuning strategies evaluated in our experiments, including Proximal FT, Soup FT, Sparse FT, LSS FT, and LP-FT. Section C gives full experimental details: computing environment, hyperparameters, dataset visualizations (Fig. 5), and dataset/model statistics (Table 6). In Section D we present additional empirical results, including PEFT distortions (Table 7) and label-shift experiments (Table 3). Section E contains full proofs of Lemma 4.3, Theorems 4.4 and 4.5, supplementary theoretical results, along with illustrative plots (Fig. 6). Finally, Section G analyzes the computational overhead of LP-FT vs. FT.

Table 5: Important notations used in the appendix

| Notation | Description |
|---|---|
| $C$ | Number of clients in federated learning |
| $m$ | Dimension of the common part of the linear head shared across clients |
| $d$ | Dimension of feature representations |
| $k$ | Dimension of the linear head |
| $B \in \mathbb{R}^{k \times d}$ | Feature extractor matrix |
| $B_* \in \mathbb{R}^{k \times d}$ | Ground-truth feature extractor matrix (with orthonormal rows) |
| $V \in \mathbb{R}^k$ | Linear head weight vector |
| $V_i^* \in \mathbb{R}^k$ | Ground-truth linear head for client $i$ |
| $V_0 \in \mathbb{R}^k$ | Initial linear head before fine-tuning |
| $V_{\text{com}}^* \in \mathbb{R}^m$ | Common component of linear heads across clients |
| $\lambda$ | Concept shift magnitude (heterogeneity parameter) |
| $\epsilon$ | Noise scale in the covariate shift model |
| $\mathcal{D}_i$ | Data distribution of client $i$ |
| $\mathcal{D}_G$ | Mixture of all clients' data distributions (global distribution) |
| $\eta$ | Learning rate used during fine-tuning |
| $\theta_L$ | Parameters of the local model |
| $\theta_G$ | Parameters of the global model |
| $\mathcal{L}_L$ | Local loss function |
| $\widehat{\mathcal{L}}_L$ | Empirical local loss |
| $\mathcal{L}_G$ | Global loss function |

## A    Related Work: Personalized Federated Learning and Fine-Tuning

**Personalized FL.** Heterogeneous federated learning (FL) refers to a decentralized training paradigm in which diverse and disparate clients, each with their own data distribution and system characteristics, collaboratively train a shared model. A standard starting point is FedAvg (McMahan et al., 2017), along with numerous improvements that can be broadly grouped into *aggregation optimization* and *local optimization.* FedNova (Wang et al., 2020) belongs to the aggregation-optimization family, normalizing and rescaling local updates to correct objective inconsistency across heterogeneous clients. Examples of local optimization include FedProx (Li et al., 2020a) and Scaffold (Karimireddy et al., 2020), where FedProx introduces a proximal ($\ell_2$) term to stabilize local training under statistical and systems heterogeneity, and Scaffold adds control variates to reduce client-drift–induced variance. Beyond these, FedBN (Li et al., 2021a) proposes local batch normalization layers to explicitly handle feature-shift non-IID settings by decoupling client-specific statistics from globally shared parameters, improving robustness to heterogeneous input distributions. More recently, several works revisit and generalize the aggregation rule itself: FedLAW (Li et al., 2023) learns aggregation weights based on client coherence rather than fixing them to data sizes, while FedAWA (Shi et al., 2025) adaptively optimizes aggregation weights using client vectors that capture the direction of local updates, both aiming to better exploit cross-client heterogeneity at the server level. However, these methods primarily focus on improving a single global model and often exhibit limited personalization capabilities, which may not adequately meet the performance requirements of different clients.

Consequently, various personalized FL approaches have been proposed, with a primary emphasis on enhancing local client performance. We can group these personalized FL strategies into clustering-based methods (Ghosh et al., 2020), transfer-learning-based personalization (Yu et al., 2020), and approaches that interpolate local and global models (Mansour et al., 2020; Deng et al., 2021). Complementary to these FL-specific methods, domain generalization techniques such as SWAD (Cha et al., 2021), which seek flat minima to improve robustness across domains, further highlight the importance of learning representations that generalize under distribution shift. Some recent FL methods (Guo et al., 2024; Son et al., 2024) similarly emphasize that standard federated training may still fail under feature shift even when label-shift issues are addressed, and propose clustering and regularization strategies to tackle diverse non-IID patterns.

In contrast to process-integrated PFL methods that modify the global training protocol (e.g., via personalized aggregation, client clustering, or feature-aligned regularization), our work focuses on a post-hoc, plug-and-play personalization layer applied after standard FedAvg training. LP-FT operates entirely in this post-hoc regime, aiming to balance personalization and global generalization without changing the underlying FL infrastructure.

**Fine-Tuning.** Fine-tuning pre-trained models has become increasingly popular with the rise of foundation models (Bommasani et al., 2021). However, fine-tuning with limited data often lead to overfitting. Several strategies can mitigate this issue, such as using optimizers that promote a flatter loss landscape (Li et al., 2018; Kaddour et al., 2022). Notably, Sharpness-Aware Minimization (SAM) (Foret et al., 2021) and Stochastic Weight Averaging (SWA) (Izmailov et al., 2018) are two popular methods that help achieve this. Additionally, a technique called *model soups* (Wortsman et al., 2022b), uses a simple greedy weight averaging approach similar to SWA, shown significant improvements in fine-tuning. An interesting perspective focuses on minimizing the linear mode connectivity barrier between the pre-trained and fine-tuned models, helping maintain consistency in decision-making mechanisms from a loss landscape perspective (Vlaar & Frankle, 2022). *Partial fine-tuning* is another common method to prevent overfitting, which involves selectively fine-tuning specific layers of the model to better adapt to variations in data distribution (Lee et al., 2023). Recent studies have introduced the concept of LP-FT (Kumar et al., 2022), highlighting potential distortions in pre-trained features and their underperformance in scenarios involving previously unseen data. Further research on LP-FT provides a deeper analysis of model adaptation (Trivedi et al., 2023), focusing on feature distortion and simplicity bias, thereby enhancing our understanding of fine-tuning mechanisms and safe model adaptation.

## B    Different Variants of Fine-tune Strategy

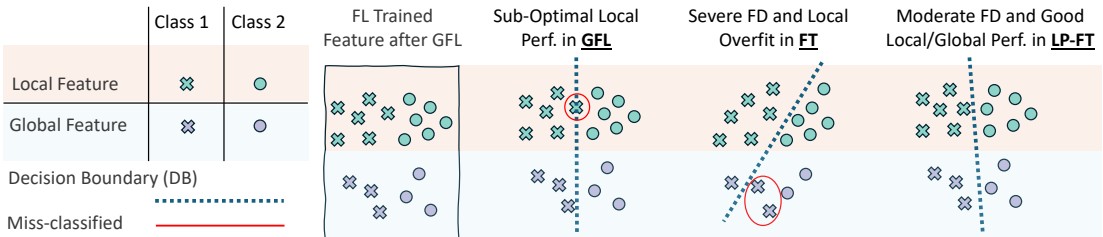

Figure 4: Illustration of federated feature distortion (FD) and decision boundaries.

This section provides an overview of the baseline techniques utilized in our study. We describe the characteristics and implementation specifics of three main fine-tuning methods: Proximal FT, Soup FT, and Sparse FT.

### B.1    Proximal FT

Proximal Fine-Tuning (Proximal FT) (Li et al., 2020b) is a method that emphasizes preserving the original knowledge of the pre-trained model while adapting it to new tasks. This technique employs proximal regularization, which penalizes large deviations from the initial model parameters during the fine-tuning

process. The primary advantage of Proximal FT is its ability to maintain the generalization capabilities of the pre-trained model, thus reducing the risk of overfitting to the new task's data. In our experiments, we used an L2 regularization term to enforce proximity between the pre-trained and fine-tuned weights, with a regularization coefficient of 0.01.

### B.2 Soup FT

Soup Fine-Tuning (Soup FT) (Wortsman et al., 2022a) is an innovative approach that leverages the concept of "model soups," where multiple fine-tuned models are combined to create a more robust final model. The key idea is to fine-tune several instances of the pre-trained model on the target task with different random initializations or data shuffling, and then average the resulting weights to form a "soup." This method aims to enhance model robustness and performance by integrating the strengths of various fine-tuning instances. For our implementation, we fine-tuned five versions of the pre-trained model and averaged their parameters to create the final Soup FT model.

### B.3 Sparse FT

Sparse Fine-Tuning (Sparse FT) (Lee et al., 2018) introduces sparsity constraints into the fine-tuning process, encouraging the model to update only a subset of its parameters. This approach aims to improve model efficiency and interpretability by ensuring that only the most relevant weights are adjusted during training. Sparse FT can be particularly beneficial for deploying models in resource-constrained environments where computational efficiency is paramount. In our experiments, we applied a gradient-based metric to the parameter prune, setting the regularization coefficient to 0.001 to achieve a balance between performance and sparsity.

### B.4 LSS FT

In the context of FL, the Local Superior Soups (LSS) methodology (Chen et al., 2024) introduces a novel approach to enhance both generalization and communication efficiency. By leveraging model interpolation-based local training, LSS encourages clients to explore a connected low-loss basin through optimizable and regularized model interpolation. This strategy not only mitigates the challenges posed by data heterogeneity but also significantly reduces the number of communication rounds required for model convergence. Empirical evaluations have demonstrated the effectiveness of LSS across various widely used FL datasets, underscoring its potential as a catalyst for the seamless adaptation of pre-trained models in federated settings. In our experiments, we fine-tuned three candidate models for model interpolation and averaged their parameters to create the final Soup FT model.

### B.5 LP-FT

Linear Probing and then Fine-Tuning (LP-FT) (Kumar et al., 2022) is a two-step transfer learning approach designed to balance in-distribution (ID) and out-of-distribution (OOD) performance. In the first step, linear probing trains only the final layer (head) while freezing the pretrained feature extractor to ensure OOD robustness. The second step fine-tunes all model parameters to improve ID accuracy while retaining the benefits of linear probing for OOD generalization. LP-FT addresses the trade-offs inherent in full fine-tuning by initializing with a well-aligned linear head, reducing feature distortion during optimization. Empirically, LP-FT demonstrates superior ID and OOD accuracy across diverse datasets.

## C Experimental Details

### C.1 Computing Environment and Hyper-parameters

All experiments in this paper are conducted on NVIDIA A40 Graphics cards using PyTorch. The Adam optimizer is employed with a learning rate of $1 \times 10^{-3}$. In FL for all datasets, the standard local model update epochs are set to 1. The communication round is set to be 100 epochs, where we validated the model

results from FL converged. Unless specified otherwise, the batch size for all benchmarks is standardized at 128. To ensure a fair comparison with various baselines, all methods initiate the FL personalized fine-tuning with models derived from the best-performing global model in terms of overall effectiveness.

## C.2  Visualization of Original Images

**Datasets and Distribution Shifts.** Figure 5 gives an overview of the benchmarks used in this study, organized by the four types of distribution shift we investigate—*feature-, input-, output-,* and *label-level.* All datasets are chosen to mirror the heterogeneity observed in real-world federated learning (FL) deployments.

- **Feature-level shift: Digit5 and DomainNet. Digit5** merges five digit domains—MNIST, SVHN, USPS, SynthDigits, and MNIST-M—whose diverse styles (e.g. grayscale scans vs. synthetic colors) induce substantial feature variations. **DomainNet** depicts everyday objects across six artistic domains (clip art, sketch, photo, *etc.*), further stressing the model's ability to transfer features across drastically different visual styles.

- **Input-level shift: CIFAR10-C and CIFAR100-C.** CIFAR10-C/100-C apply 50 corruption types (Gaussian noise, motion blur, pixelation, brightness, contrast, . . . ) to the original CIFAR images, emulating degradations caused by hardware or environment. These pixel-level perturbations leave semantics intact while challenging a model's robustness to distorted inputs—crucial for FL settings such as mobile sensing and autonomous driving.

- **Output-level shift: CheXpert and CelebA.** In **CheXpert**, we focus on two labels (Edema, No Finding) and partition clients to introduce demographic biases (e.g. male patients predominating in Edema, female in No Finding). **CelebA** clients are built by correlating gender and hair-color attributes (e.g. "blonde-haired females" vs. "non-blonde males"), yielding skewed label distributions that test a model's resilience to demographic imbalance.

- **Label-level shift: CIFAR10 (Dirichlet-$\alpha$=0.1).** We partition the clean CIFAR-10 dataset among 20 clients using a Dirichlet prior with $\alpha = 0.1$, producing highly imbalanced class proportions. Such label shift typifies FL scenarios where each client serves a niche population—e.g. hospitals specializing in different diseases—violating the IID assumption of standard aggregation rules.

## C.3  Detailed Dataset and Model Information

**Table. 6** provides a visual overview of the datasets used in this study, categorized by their levels of transformation and data heterogeneity. The table is divided into four sections, corresponding to feature-level, input-level, output-level, and label shift settings:

*Feature-Level Shift (Digit5 and DomainNet):* The `Digit5` dataset, which consists of digit images collected from five distinct domains, including MNIST, SVHN, USPS, SynthDigits, and MNIST-M. These images exhibit a variety of styles, such as handwritten digits, digits rendered in different fonts, and textured representations, demonstrating substantial visual heterogeneity. The `DomainNet` dataset is a large-scale collection featuring objects and scenes from six domains, including styles like clip art, sketches, and realistic photographs.

*Input-Level Shift (`CIFAR10-C` and `CIFAR100-C`):* This type of distribution shift includes corrupted versions of CIFAR-10 and CIFAR-100 datasets. The **CIFAR10-C** dataset applies 50 types of corruptions, such as noise, blur, and distortions, to evaluate model robustness under various degradation conditions. Similarly, **CIFAR100-C** extends the CIFAR-100 dataset by introducing the same set of corruptions, enabling robustness evaluation on a larger and more diverse set of categories.

*Output-Level Shift (CheXpert and CelebA):* The **CheXpert** dataset, a widely used medical imaging dataset labeled for 14 common chest conditions. In this study, the Edema and No Finding labels are grouped, and spurious correlations are introduced at the client level, where attributes (i.e. gender in our client splitting) are disproportionately represented (i.e., 90% of label 1 examples in a client are a certain attribute). The **CelebA** dataset, which includes over 200,000 celebrity faces annotated with 40 attributes. Client splitting

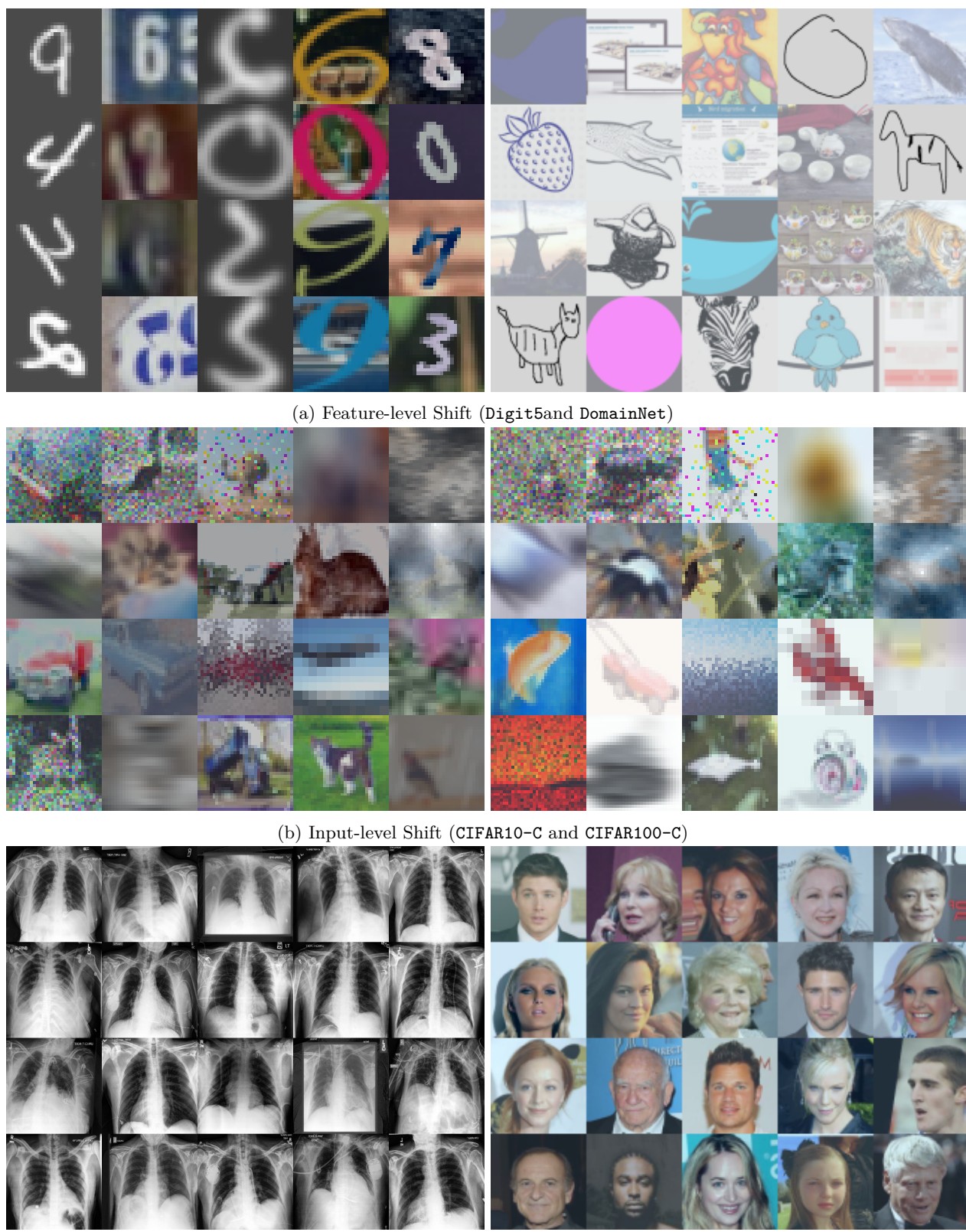

(a) Feature-level Shift (`Digit5`and `DomainNet`)

(b) Input-level Shift (`CIFAR10-C` and `CIFAR100-C`)

(c) Output-level Shift (`ChexPert` and `CelebA`)

Figure 5: Visualization of the original datasets used in the paper.

| Data Heterogeneity Type | Dataset | Model | Description | Clients | Classes |
|---|---|---|---|---|---|
| **Feature-Level Shift** | Digit5 | ResNet18 | A collection of digit images from five domains, used for domain adaptation and digit recognition tasks (Huang et al., 2023). Datasets include MNIST, SVHN, USPS, SynthDigits, and MNIST-M. | 5 | 10 |
| | DomainNet | ResNet50 | A large-scale dataset of images from six distinct domains for multi-source domain adaptation (Peng et al., 2019b). Preprocessing follows the strategy in FedBN (Li et al., 2021b). | 6 | 10 |
| **Input-Level Shift** | CIFAR10-C | ResNet18 | A corruption benchmark dataset for CIFAR-10 (Hendrycks & Dietterich, 2019), augmented with 30 additional corruption types (Mintun et al., 2021) and one extra type (Chen et al., 2021), totaling 50 corruption types (including the original 19 types of corruption from CIFAR-10-C, plus 30 additional corruption types and one extra type, resulting in a total of 50 distinct corruption types). | 50 | 10 |
| | CIFAR100-C | ResNet18 | An extension of CIFAR-100 with common corruptions, following the same strategy as CIFAR-10-C. | 50 | 100 |
| **Output-Level Shift** | CheXpert | ResNet50 | A chest radiograph dataset labeled for 14 common chest conditions (Irvin et al., 2019). Edema and No Finding labels are grouped as described in (Jin et al., 2024). Clients are spuriously correlated with the attribute gender (e.g., 90% of label 1 examples in a client are male). | 2 | 2 |
| | CelebA | ResNet50 | Over 200,000 celebrity images with 40 attributes (Liu et al., 2015). Client splitting follows the same strategy as CheXpert, with attributes: male, female, blonde hair, and non-blonde hair. | 4 | 2 |
| **Label Shift** | CIFAR10 | ResNet18 | A benchmark dataset with label shift induced via Dirichlet distribution ($\alpha = 0.1$), distributed across 20 clients (Krizhevsky et al., 2009). | 20 | 10 |

Table 6: Detailed information about the datasets used in the study. For all the settings above, each client has an individual data distribution, while different clients prohibits the corresponding shifts to ensure the non-IID nature required for heterogeneous FL. Feature-level shift, also referred to as subgroup shift, and input-level shift, corresponding to image corruption, are categorized as covariate shift. Output-level shift, representing spurious correlations in our setting, is categorized as concept shift.

follows the same strategy as CheXpert, with attributes such as male, female, blonde hair, and non-blonde hair used to create spurious correlations across clients.

*Label Shift (CIFAR10):* The final one focuses on label shift in the **CIFAR10** dataset. This setting simulates non-IID conditions by inducing label distributions across clients using a Dirichlet distribution with $\alpha = 0.1$. This creates significant variations in class distributions among the 20 clients, mimicking real-world federated learning scenarios where data availability across clients is inherently imbalanced.

# D   Further Empirical Results

## D.1   Exploration of Parameter-Efficient Fine-Tuning Under Federated Feature Distortion

**Setup.**   In this study, we adapted two widely recognized Parameter-Efficient Fine-Tuning (PEFT) methods: LoRA (Hu et al., 2022) and Adapter (Houlsby et al., 2019). Both methods were fine-tuned with carefully

| PEFT Method | Local | Global | Avg. |
|---|---|---|---|
| **LoRA (lr=1e-3)** | 41.54 | 26.75 | 26.87 |
| **LoRA (lr=1e-4)** | 42.01 | 26.41 | 27.18 |
| **Adapter (lr=1e-3)** | 48.35 | 39.28 | 38.08 |
| **Adapter (lr=1e-4)** | 49.07 | 39.83 | 38.36 |
| **LP-FT** | 68.50 | 57.52 | 53.52 |

Table 7: Different PEFT compared on `DomainNet` with ViT.

tuned learning rates on the ViT, as detailed in Tab. 7. These configurations allowed us to assess the impact of different fine-tuning strategies on federated features. The performance of each method was measured in terms of local, global, and average accuracy.

**Result.** We emphasize that this PEFT analysis is not part of the core LP-FT methodology nor a competing approach. Instead, it serves as an auxiliary and informative experiment aimed at understanding whether modern parameter-efficient tuning mechanisms can implicitly mitigate the federated feature distortion phenomenon identified in this work.

The effectiveness of bias-tuning naturally raises the question of whether other PEFT methods—originally developed for fine-tuning large models in centralized settings—behave similarly in the personalized FL setting. To investigate this, we compare the local and global performance of LoRA and Adapter against LP-FT.

Our findings reveal that while PEFT methods can achieve strong local accuracy, their global accuracy drops substantially, indicating that they distort the shared federated features. Consequently, PEFT methods still fall short of LP-FT in both global and average accuracy, demonstrating that parameter-efficiency alone does not resolve federated feature distortion.

These observations highlight an important implication of our work: techniques designed for centralized model adaptation do not directly translate into personalized FL, where preserving cross-client feature coherence is essential. PEFT methods restrict *which* parameters are adapted for efficiency, whereas LP-FT controls *how* fine-tuning unfolds temporally to mitigate distortion.

Taken together, this auxiliary experiment suggests a promising future direction: developing PEFT approaches that explicitly account for federated feature distortion, while reinforcing that LP-FT fills a gap unmet by existing fine-tuning strategies.

### D.2 Comparison with Process-Integrated PFL Methods

Table 8: Comparison with Process-Integrated PFL Methods on CelebA

| PFL Method | Local Acc. (↑) | Global Acc. (↑) | Avg. Acc. (↑) |
|---|---|---|---|
| FedBN (Li et al., 2021) | 88.68 | 61.35 | 61.00 |
| PerAvg (Fallah et al., 2020) | 87.06 | 67.26 | 66.66 |
| FedNova (Wang et al., 2020) | 88.68 | 54.26 | 53.41 |
| FedRep (Collins et al., 2021) | 86.94 | 52.99 | 52.57 |
| FedSoup (Chen et al., 2023) | 90.30 | 75.62 | 75.21 |
| pFedFDA (McLaughlin et al., 2024) | 90.41 | 76.56 | 76.63 |
| FedL2G (Zhang et al., 2024) | 92.46 | 79.27 | 78.82 |
| **LP-FT (ours)** | **93.03** | **82.46** | **82.17** |

To further contextualize the effectiveness of our post-hoc PFT approach, we compare FedAvg + LP-FT against several representative *process-integrated* personalized federated learning (PFL) methods on the CelebA dataset. Unlike our plug-and-play setting, these baselines modify the global training pipeline through personalized aggregation, client clustering, or feature-adaptive updates, and therefore represent a different but highly

competitive regime of PFL. All methods are evaluated under the same CelebA partitioning and backbone architecture, and we report the local accuracy on each client's own test distribution, the global accuracy on the aggregated test distribution, and their average.

As shown in Table 8, LP-FT achieves the best performance across all three metrics, surpassing even the strongest process-integrated PFL methods in both local and global accuracy. This result is noteworthy because LP-FT operates *exclusively* as a post-hoc personalization layer on top of a standard FedAvg model, without altering the server–client protocol, communication pattern, or global optimization objective. The fact that such a simple, deployment-friendly PFT strategy can match or outperform substantially more complex PFL pipelines supports our central claim: LP-FT constitutes a strong and practical baseline that simultaneously preserves global generalization and delivers high-quality personalization, while remaining compatible with existing federated learning infrastructures.

## E   Proofs and Supplementary Results

*Proof of Lemma 4.3.* We want to analyze the Fine-Tuning (FT) method, focusing on the effect of initial parameters. We perform one pass through the entire dataset to simulate the complete fine-tuning process. Consider the Mean Squared Error (MSE) loss function with parameters $V$ and $B$, where $B$ is represented as follows:

$$
B = \begin{bmatrix} b_1^T \\ \vdots \\ b_m^T \\ b_{m+1}^T \\ \vdots \\ b_{m+C}^T \end{bmatrix},
$$

where $b_i^T \in \mathbb{R}^{1 \times d}$ denotes the $i$-th row of matrix $B$, and $m + C = k$.

To apply one step of gradient descent, we need to compute the gradient of the loss function with respect to $V$, $b_1$, $b_2$, ..., $b_{m+C}$, and then perform one update step.

W.L.O.G. we assume the local client is client 1. We define the local loss function as follows:

$$
\mathcal{L}_L(V, B) = \mathbb{E}_{x \sim \mathcal{D}_1} \left[ \frac{1}{2} (V^T B x - V_1^{*T} B_* x)^2 \right],
$$

where $\mathcal{D}_1$ is the data distribution for client 1.

Now, let $(\mathbf{X}_1, \mathbf{Y}_1)$ represent the local dataset of client 1, consisting of $n_1$ data points $\{(x_{1j}, y_{1j})\}_{j=1}^{n_1}$. We aim to calculate the gradient of the empirical loss function with respect to the parameters. The empirical loss function is given by:

$$
\widehat{\mathcal{L}_L}(V, B) = \frac{1}{n_1} \sum_{j=1}^{n_1} \left[ \frac{1}{2} (V^T B x_{1j} - V_1^{*T} B_* x_{1j})^2 \right].
$$

In practice, we take the gradient of this empirical loss function with respect to the parameters $V$, $b_1$, $b_2$, ..., $b_{m+C}$. However, since we are particularly interested in computing the expectation $\mathbb{E}[b_j^{FT}]$, we evaluate the expected value of the gradients using one pass through the whole dataset as follows:

$$
\mathbb{E}\left[ \frac{\partial \widehat{\mathcal{L}_L}}{\partial V} \bigg|_{\substack{V=V_0 \\ B=B_*}} \right] = \mathbb{E}\left[ \frac{\partial}{\partial V} \left( \frac{1}{n_1} \sum_{j=1}^{n_1} \frac{1}{2} \left( V^T B x_{1j} - V_1^{*T} B_* x_{1j} \right)^2 \right) \bigg|_{\substack{V=V_0 \\ B=B_*}} \right]
$$

$$
= \frac{1}{n_1} \sum_{j=1}^{n_1} \mathbb{E}\left[ (V_0^T B_* x_{1j} - V_1^{*T} B_* x_{1j}) x_{1j}^T B_*^T \right]
$$

$$
= \mathbb{E}_{x \sim \mathcal{D}_1} \left[ (V_0^T B_* x - V_1^{*T} B_* x) x^T B_*^T \right].
$$

This follows because the dataset is drawn i.i.d. from the population, making the empirical gradient an unbiased estimate of the true gradient. Therefore, let $V_0 = \begin{bmatrix} V_{com}^* {}^T & \mathbf{0}^T \end{bmatrix}^T$. It follows that:

$$\mathbb{E}\left[\left.\frac{\partial L}{\partial V}\right|_{\substack{V=V_0 \\ B=B_*}}\right] = \mathbb{E}_{x \sim \mathcal{D}_1}\left[(V_0^T B_* x - V_1^* {}^T B_* x)x^T B_*^T\right]$$

$$= \mathbb{E}_{x \sim \mathcal{D}_1}\left[((V_0 - V_1^*)^T B_* x)x^T B_*^T\right]$$

$$= (V_0 - V_1^*)^T B_* \left(\mathbb{E}_{x \sim \mathcal{D}_1}\left[xx^T\right]\right)B_*^T$$

$$= (V_0 - V_1^*)^T B_* B_*^T \qquad \text{(second moment is identity)}$$

$$= (V_0 - V_1^*)^T. \qquad \text{($B_*$ has orthonormal rows)}$$

Let $B_* = \begin{bmatrix} b_1^* {}^T \\ \vdots \\ b_m^* {}^T \\ b_{m+1}^* {}^T \\ \vdots \\ b_{m+C}^* {}^T \end{bmatrix}$. Then, similarly, it can be shown that:

$$\mathbb{E}\left[\left.\frac{\partial L}{\partial b_j}\right|_{\substack{V=V_0 \\ B=B_*}}\right] = \mathbb{E}_{x \sim \mathcal{D}_1}\left[(V_0^T B_* x - V_1^* {}^T B_* x)(V_0)_j x^T\right]$$

$$= \mathbb{E}_{x \sim \mathcal{D}_1}\left[(V_0)_j\left((V_0 - V_1^*)^T B_* x\right)x^T\right]$$

$$= (V_0)_j(V_0 - V_1^*)^T B_* \left(\mathbb{E}_{x \sim \mathcal{D}_1}\left[xx^T\right]\right)$$

$$= (V_0)_j(V_0 - V_1^*)^T B_*. \qquad \text{(second moment is identity)}$$

Here, $(V_0)_j$ is the $j$-th element of the vector $V_0$. For learning rate $\eta$, one step of gradient descent is:

$$V_{FT} = V_0 - \eta\left(\left.\frac{\partial L}{\partial V}\right|_{\substack{V=V_0 \\ B=B_*}}\right)^T$$

$$b_j^{FT} = b_j^* - \eta\left(\left.\frac{\partial L}{\partial b_j}\right|_{\substack{V=V_0 \\ B_0=B_*}}\right)^T.$$

These two equations can be further refined as:

$$\mathbb{E}[V_{FT}] = \begin{bmatrix} V_{com}^* {}^T & \mathbf{0}^T \end{bmatrix}^T - \eta(V_0 - V_1^*) = \begin{bmatrix} V_{com}^* {}^T & \eta\lambda e_1^T \end{bmatrix}^T$$

$$\mathbb{E}[b_j^{FT}] = \mathbb{E}\left[b_j^* - \eta\left(\left.\frac{\partial L}{\partial b_j}\right|_{\substack{V=V_0 \\ B_0=B_*}}\right)^T\right] = b_j^* - \eta\left((V_0)_j(V_0 - V_1^*)^T B_*\right)^T$$

$$= b_j^* - \eta\lambda\left((V_0)_j\begin{bmatrix} \mathbf{0}^T & -e_1^T \end{bmatrix} B_*\right)^T = b_j^* + \eta\lambda(V_0)_j b_{m+1}^*.$$

Therefore, we have:

$$\mathbb{E}[B_{FT}] = \begin{bmatrix} b_1^* {}^T + \eta\lambda(V_0)_1 b_{m+1}^* {}^T \\ \vdots \\ b_m^* {}^T + \eta\lambda(V_0)_m b_{m+1}^* {}^T \\ b_{m+1}^* {}^T + \eta\lambda(V_0)_{m+1} b_{m+1}^* {}^T \\ \vdots \\ b_{m+C}^* {}^T + \eta\lambda(V_0)_{m+C} b_{m+1}^* {}^T \end{bmatrix}$$

$$= \begin{bmatrix} b_1^{*T} + \eta\lambda(V_0)_1 {b_{m+1}^*}^T \\ \vdots \\ {b_m^*}^T + \eta\lambda(V_0)_m {b_{m+1}^*}^T \\ {b_{m+1}^*}^T \\ \vdots \\ {b_{m+C}^*}^T \end{bmatrix} = \begin{bmatrix} b_1^{*T} + \eta\lambda(V_{com}^*)_1 {b_{m+1}^*}^T \\ \vdots \\ {b_m^*}^T + \eta\lambda(V_{com}^*)_m {b_{m+1}^*}^T \\ {b_{m+1}^*}^T \\ \vdots \\ {b_{m+C}^*}^T \end{bmatrix}.$$

Similarly, if the fine-tuning is done over the data of client $i$, we would have:

$$\mathbb{E}\big[b_j^{FT}\big] = b_j^* + \eta\lambda(V_0)_j b_{m+i}^*,$$

which concludes the proof. $\qquad\square$

*Proof of Theorem 4.4.* We assume that the pre-trained model perfectly captures the feature extractor matrix $B_*$, and its linear head represents the common part shared across all clients, excluding any client-specific components of the ground-truth function. Thus, $B_0 = B_*$ and $V_0 = \begin{bmatrix} {V_{com}^*}^T & \mathbf{0} \end{bmatrix}^T$. In this setting, we analyze the effects of LP-FT and FT on the model parameters. For both LP-FT and FT, we determine the parameters after fine-tuning, compute the global loss, and then compare these global losses.

W.L.O.G. we assume that we are doing the fine-tuning over the local data of client 1. First, we study FT.

It can be shown that:

$$
\begin{aligned}
\mathcal{L}_G(V_{FT}, B_{FT}) &= \frac{1}{C} \sum_{i \in [C]} \mathbb{E}_{x \sim \mathcal{D}_i} \left[ \frac{1}{2} (V_{FT}^T B_{FT} x - V_i^{*T} B_* x)^2 \right] \\
&= \frac{1}{2C} \sum_{i \in [C]} \mathbb{E}_{x \sim \mathcal{D}_i} \left[ (B_{FT}^T V_{FT} - B_*^T V_i^*)^T x x^T (B_{FT}^T V_{FT} - B_*^T V_i^*) \right] \\
&= \frac{1}{2C} \sum_{i \in [C]} (B_{FT}^T V_{FT} - B_*^T V_i^*)^T \left[ \mathbb{E}_{x \sim \mathcal{D}_i} \big[ x x^T \big] \right] (B_{FT}^T V_{FT} - B_*^T V_i^*) \\
&= \frac{1}{2C} \sum_{i \in [C]} (B_{FT}^T V_{FT} - B_*^T V_i^*)^T (B_{FT}^T V_{FT} - B_*^T V_i^*) \qquad \text{(second moment is } I_d) \\
&= \frac{1}{2C} \sum_{i \in [C]} \big\| (B_{FT}^T V_{FT} - B_*^T V_i^*) \big\|_2^2. \qquad\qquad\qquad\qquad\qquad (1)
\end{aligned}
$$

We have:

$$B_*^T V_i^* = \sum_{j=1}^m (V_{com}^*)_j b_j^* + \lambda b_{m+i}^*$$

$$\mathbb{E}\left[ B_{FT}^T V_{FT} \right] = \sum_{j=1}^m (V_{com}^*)_j b_j^* + \sum_{j=1}^m \eta\lambda(V_{com}^*)_j{}^2 b_{m+1}^* + \eta\lambda b_{m+1}^*.$$

Therefore, we can obtain:

$$\mathbb{E}\left[ B_{FT}^T V_{FT} - B_*^T V_i^* \right] = \lambda\Big( \sum_{j=1}^m \eta(V_{com}^*)_j{}^2 b_{m+1}^* + \eta b_{m+1}^* - b_{m+i}^* \Big).$$

For $i \neq 1$, we have:

$$
\begin{aligned}
&\mathbb{E}\big[ (B_{FT}^T V_{FT} - B_*^T V_i^*)^T (B_{FT}^T V_{FT} - B_*^T V_i^*) \big] \\
&\qquad = \lambda^2 \big( \sum_{j=1}^m \eta(V_{com}^*)_j{}^2 b_{m+1}^* + \eta b_{m+1}^* - b_{m+i}^* \big)^T \big( \sum_{j=1}^m \eta(V_{com}^*)_j{}^2 b_{m+1}^* + \eta b_{m+1}^* - b_{m+i}^* \big)
\end{aligned}
$$

$$= \lambda^2 \left( \left( \eta + \eta \sum_{j=1}^{m} (V_{com}^*)_j^2 \right)^2 + 1 \right). \qquad \text{(rows of } B_* \text{ are orthonormal)}$$

For $i = 1$, we have:

$$\mathbb{E}\left[ (B_{FT}^T V_{FT} - B_*^T V_i^*)^T (B_{FT}^T V_{FT} - B_*^T V_i^*) \right]$$

$$= \lambda^2 \left( \sum_{j=1}^{m} \eta (V_{com}^*)_j^2 b_{m+1}^* + \eta b_{m+1}^* - b_{m+i}^* \right)^T \left( \sum_{j=1}^{m} \eta (V_{com}^*)_j^2 b_{m+1}^* + \eta b_{m+1}^* - b_{m+i}^* \right)$$

$$= \lambda^2 \left( \eta + \eta \sum_{j=1}^{m} (V_{com}^*)_j^2 - 1 \right)^2. \qquad \text{(rows of } B_* \text{ are orthonormal)}$$

Combining these with (1), we can conclude:

$$\mathcal{L}_G(V_{FT}, B_{FT}) = \mathbb{E}\left[ \frac{1}{2C} \sum_{i \in [C]} \| (B_{FT}^T V_{FT} - B_*^T V_i^*) \|_2^2 \right]$$

$$= \frac{\lambda^2}{2C} \left( \left( \eta + \eta \sum_{j=1}^{m} (V_{com}^*)_j^2 - 1 \right)^2 + (C-1) \left( \left( \eta + \eta \sum_{j=1}^{m} (V_{com}^*)_j^2 \right)^2 + 1 \right) \right). \qquad (2)$$

We can follow the same procedure for LP-FT as follows:

LP step starts from $B_0 = B_*$ and $V_0 = \begin{bmatrix} V_{com}^*{}^T & \mathbf{0} \end{bmatrix}^T$. From proof of Lemma 4.3, we know that $\mathbb{E}\left[ \frac{\partial L}{\partial V} \big|_{\substack{V=V_0 \\ B=B_*}} \right] = (V_0 - V_1^*)^T$. Therefore, after one iteration of LP, we have:

$$V_{LP} = V_0 - \eta \left( \frac{\partial L}{\partial V} \Big|_{\substack{V=V_0 \\ B=B_*}} \right)^T$$

After $I$ iterations we have:

$$\mathbb{E}\left[ V_{LP} \right] = \begin{bmatrix} V_{com}^*{}^\top & \lambda \alpha e_1^\top \end{bmatrix}^\top \qquad \text{for} \quad \alpha = (1 - (1 - \eta)^I)$$

Let the initial parameters after one step of LP and before the FT step be

$$B_{LP} = B_*, \qquad \mathbb{E}\left[ V_{LP} \right] = \begin{bmatrix} V_{com}^*{}^\top & \lambda \alpha e_1^\top \end{bmatrix}^\top, \quad \alpha \in [0, 1].$$

Assume the first client ($i = 1$) runs a single full-batch gradient–descent step with learning–rate $\eta$ on its local MSE loss. Let $V_{LPFT}$ and $B_{LPFT}$ be the resulting linear head and feature-extractor matrix and $(b_j^{LPFT})^\top$ its $j$-th row after the final FT step. Then

$$\mathbb{E}\left[ V_{LPFT} \right] = \begin{bmatrix} V_{com}^*{}^T & \lambda (\alpha + \eta(1 - \alpha)) e_1^T \end{bmatrix}^T$$

$$\mathbb{E}\left[ b_j^{LPFT} \right] = \mathbb{E}\left[ b_j^* - \eta \left( \frac{\partial L}{\partial b_j} \Big|_{\substack{V=V_{LP} \\ B_{LP}=B_*}} \right)^T \right] = b_j^* - \eta \left( (V_{LP})_j (V_{LP} - V_1^*)^T B_* \right)^T$$

$$= b_j^* - \eta \lambda (1 - \alpha) \left( (V_{LP})_j \begin{bmatrix} \mathbf{0}^T & -e_1^T \end{bmatrix} B_* \right)^T = b_j^* + \eta \lambda (1 - \alpha) (V_{LP})_j b_{m+1}^*.$$

This steps can be directly obtained exactly as we derived $\mathbb{E}\left[ V_{FT} \right]$ and $\mathbb{E}\left[ b_j^{FT} \right]$ for FT case.

Similar to computing $\mathcal{L}_G(V_{FT}, B_{FT})$, we find $\mathcal{L}_G(V_{LPFT}, B_{LPFT})$ as follows:

Writing $S := \sum_{j=1}^m (V_{\text{com}}^*)_j^2 = \|V_{\text{com}}^*\|_2^2$ and

$$A_\alpha := \eta(1-\alpha)S + \big[\alpha + \eta(1-\alpha)\big]\big[1 + \eta\lambda^2\alpha(1-\alpha)\big],$$

the expected global loss

$$\mathcal{L}_G(V_{LPFT}, B_{LPFT}) := \frac{1}{C}\sum_{i=1}^C \mathbb{E}_{x\sim\mathcal{D}_i}\Big[\tfrac{1}{2}\big(V_{LPFT}^\top B_{LPFT}x - V_i^{*\top}B_*x\big)^2\Big]$$

satisfies

$$\mathcal{L}_G(V_{LPFT}, B_{LPFT}) = \frac{\lambda^2}{2C}\Big[(A_\alpha - 1)^2 + (C-1)A_\alpha^2\Big].$$

this is because the LP-FT loss can be obtained similar to FT loss as in (1),

$$\mathcal{L}_G(V_{LPFT}, B_{LPFT}) = \mathbb{E}\Big[\frac{1}{2C}\sum_{i=1}^C \|B_{LPFT}^\top V_{LPFT} - B_*^\top V_i^*\|_2^2\Big].$$

First, it can be shown that:

$$\mathbb{E}\Big[B_{LPFT}^\top V_{LPFT}\Big] = \sum_{j=1}^m (V_{\text{com}}^*)_j b_j^* + \lambda A_\alpha b_{m+1}^*.$$

For each client $i \in [C]$, $B_*^\top V_i^* = \sum_{j=1}^m (V_{\text{com}}^*)_j b_j^* + \lambda b_{m+i}^*$. Therefore, defining

$$\Delta_i := \mathbb{E}\Big[B_{LPFT}^\top V_{LPFT} - B_*^\top V_i^*\Big],$$

and noting the orthonormality of the rows of $B_*$, we obtain

$$\|\Delta_i\|_2^2 = \begin{cases} \lambda^2(A_\alpha - 1)^2, & i = 1, \\ \lambda^2 A_\alpha^2, & i \neq 1. \end{cases}$$

Substituting the above norms in the global loss,

$$\mathcal{L}_G(V_{LPFT}, B_{LPFT}) = \frac{\lambda^2}{2C}\Big[(A_\alpha - 1)^2 + (C-1)A_\alpha^2\Big].$$

which is the stated result.

Since we consider convergence for the first (linear probing) step, at convergence we have $\alpha \to 1$, and thus $A_\alpha \to 1$. Therefore, we obtain

$$\mathcal{L}_G(V_{LPFT}, B_{LPFT}) = \frac{\lambda^2}{2C}\big[(A_\alpha - 1)^2 + (C-1)A_\alpha^2\big] \xrightarrow[\alpha\to 1]{} \frac{\lambda^2}{2C}(C-1).$$

In contrast, from (2), we have:

$$\mathcal{L}_G(V_{FT}, B_{FT}) = \frac{\lambda^2}{2C}\left(\Big(\eta + \eta\sum_{j=1}^m (V_{com}^*)_j^2 - 1\Big)^2 + (C-1)\Big(\big(\eta + \eta\sum_{j=1}^m (V_{com}^*)_j^2\big)^2 + 1\Big)\right).$$

Since $\lambda, C \geq 0$, $\big(\eta + \eta\sum_{j=1}^m (V_{com}^*)_j^2 - 1\big)^2 \geq 0$ and $\big(\eta + \eta\sum_{j=1}^m (V_{com}^*)_j^2\big)^2 \geq 0$, each term in $\mathcal{L}_G(V_{FT}, B_{FT})$ exceeds its counterpart in $\mathcal{L}_G(V_{LPFT}, B_{LPFT})$. Hence,

$$\mathcal{L}_G(V_{LPFT}, B_{LPFT}) \leq \mathcal{L}_G(V_{FT}, B_{FT}),$$

which concludes the proof.

$\square$

**Corollary E.1** (Local performance under concept shift). *Under the assumptions of Theorem 4.4, for any fixed client $i \in [C]$, the corresponding local loss satisfies*

$$\mathcal{L}_L^{(i)}(V_{LPFT}, B_{LPFT}) \leq \mathcal{L}_L^{(i)}(V_{FT}, B_{FT}),$$

*where $\mathcal{L}_L^{(i)}$ denotes the local loss evaluated on client $i$'s data.*

*Proof.* Recall that the global loss $\mathcal{L}_G$ is defined as the average of client-specific local losses. The proof of Theorem 4.4 derives an explicit closed-form expression for $\mathcal{L}_G(V_{FT}, B_{FT})$, from which the contribution of each individual client can be isolated. Fixing a client $i$ and extracting the corresponding term yields

$$\mathcal{L}_L^{(i)}(V_{FT}, B_{FT}) = \frac{\lambda^2}{2}\left(\eta + \eta \sum_{j=1}^{m} (V_{com}^*)_j{}^2 - 1\right)^2,$$

while for LP-FT the local loss takes the form

$$\mathcal{L}_L^{(i)}(V_{LPFT}, B_{LPFT}) = \frac{\lambda^2}{2}(A_\alpha - 1)^2.$$

The same ordering established at the global level in Theorem 4.4 immediately carries over to each individual client. $\qquad\square$

*Proof of Theorem 4.5.* W.L.O.G. we assume that the local fine-tuning is performed on the data of the first client. Initially, one step of linear probing is conducted with the fixed feature extractor $B_*$. After this step, the linear head $V_{LP}$ will converge to $V_1^*$. This is because we know that:

$$\arg\min_v \left\| \mathbf{X_1} B_0^\top v - \mathbf{X_1} B_*^\top v_* \right\|_2^2 = \left( B_0 \mathbf{X_1}^\top \mathbf{X_1} B_0^\top \right)^{-1} B_0 \mathbf{X_1}^\top \mathbf{X_1} B_*^\top v_*,$$

where $\mathbf{X_1}$ is the $n \times d$ matrix including data of $n$ individuals. Since the fine-tuning is on the data of the client 1 (local data), we have:

$$V_{LP} = \left( B_0 \mathbf{X_1}^\top \mathbf{X_1} B_0^\top \right)^{-1} B_0 \mathbf{X_1}^\top \mathbf{X_1} B_*^\top V_1^*.$$

Therefore, we have:

$$\begin{aligned}
V_{LP} &= \left( B_0 \mathbf{X_1}^\top \mathbf{X_1} B_0^\top \right)^{-1} B_0 \mathbf{X_1}^\top \mathbf{X_1} B_*^\top V_1^* \\
&= \left( B_* \mathbf{X_1}^\top \mathbf{X_1} B_*^\top \right)^{-1} B_* \mathbf{X_1}^\top \mathbf{X_1} B_*^\top V_1^* \\
&= V_1^*.
\end{aligned}$$

The argument parallels the opening of the proof of Theorem 4.4, where we simulate $I$ iterations of LP. Since at the beginning of the fine-tuning (FT) step, we have the perfect $B_*$ and $V_1^*$ for the local client 1, and FT is performed on the data of the same client, we can conclude that after one step of FT following LP, the parameters will remain unchanged. Specifically, we have $V_{LPFT} = V_1^* = \begin{bmatrix} V_{com}^*{}^T & \lambda e_1^T \end{bmatrix}^T$ and $B_{LPFT} = B_*$.

For the performance on the global data, we have:

$$\begin{aligned}
\mathcal{L}_G(V_{LPFT}, B_{LPFT}) &= \frac{1}{C} \sum_{i \in [C]} \mathbb{E}_{x \sim \mathcal{D}_i} \left[ \frac{1}{2}(V_{LPFT}^T B_{LPFT} x - V_i^{*T} B_* x)^2 \right] \\
&= \frac{1}{C} \sum_{i \in [C]} \mathbb{E}_{x \sim \mathcal{D}_i} \left[ \frac{1}{2}(V_1^{*T} B_* x - V_i^{*T} B_* x)^2 \right] \\
&= \frac{1}{2C} \sum_{i \in [C]} \mathbb{E}_{x \sim \mathcal{D}_i} \left[ (B_*^T V_1^* - B_*^T V_i^*)^T x x^T (B_*^T V_1^* - B_*^T V_i^*) \right]
\end{aligned}$$

$$= \frac{1}{2C} \sum_{i \in [C]} \left[ (B_*^T V_1^* - B_*^T V_i^*)^T \mathbb{E}_{x \sim \mathcal{D}_i}[xx^T](B_*^T V_1^* - B_*^T V_i^*) \right]$$

$$= \frac{1}{2C} \sum_{i \in [C]} \left[ (V_1^* - V_i^*)^T B_* \left( \mathbb{E}_{x \sim \mathcal{D}_i}[xx^T] \right) B_*^T (V_1^* - V_i^*) \right]$$

$$= \frac{1}{2C} \sum_{i \in [C]} \left[ (V_1^* - V_i^*)^T B_* \left( \mathbb{E}_{z \sim \mathcal{N}(0, I_d)} \left[ (e_i + \epsilon z)(e_i + \epsilon z)^T \right] \right) B_*^T (V_1^* - V_i^*) \right]$$

$$= \frac{1}{2C} \sum_{i \in [C]} \left[ (V_1^* - V_i^*)^T B_* \left( e_i e_i^T + \epsilon^2 \mathbb{E}_{z \sim \mathcal{N}(0, I_d)} \left[ zz^T \right] \right) B_*^T (V_1^* - V_i^*) \right]$$

$$= \frac{1}{2C} \sum_{i \in [C]} \left[ (V_1^* - V_i^*)^T B_* \left( e_i e_i^T + \epsilon^2 I_d \right) B_*^T (V_1^* - V_i^*) \right]$$

$$= \frac{1}{2C} \sum_{i \in [C]} \left[ (V_1^* - V_i^*)^T B_* \left( e_i e_i^T \right) B_*^T (V_1^* - V_i^*) \right]$$

$$\quad + \frac{1}{2C} \sum_{i \in [C]} \left[ (V_1^* - V_i^*)^T B_* \left( \epsilon^2 I_d \right) B_*^T (V_1^* - V_i^*) \right]$$

$$= \frac{1}{2C} \sum_{i \in [C]} \left[ (V_1^* - V_i^*)^T (B_*)_{:,i} (B_*)_{:,i}^T (V_1^* - V_i^*) \right] \qquad ((B_*)_{:,i} \text{ } i\text{-th column of } B_*)$$

$$\quad + \frac{1}{2C} \sum_{i \in [C]} \left[ \epsilon^2 (V_1^* - V_i^*)^T (V_1^* - V_i^*) \right] \qquad (B_* \text{ has orthonormal rows})$$

$$= \frac{\lambda^2}{2C} \sum_{i \in [C]} \left[ \left( (B_*)_{m+1,i} - (B_*)_{m+i,i} \right)^2 \right] + \frac{1}{2C} \epsilon^2 \sum_{i \in [C]} \left[ \left\| (V_1^* - V_i^*) \right\|_2^2 \right]$$

$$= \frac{\lambda^2}{2C} \sum_{i \in [C]} \left[ \left( (B_*)_{m+1,i} - (B_*)_{m+i,i} \right)^2 \right] + \frac{\lambda^2 (C-1)}{C} \epsilon^2. \tag{3}$$

We want to analyze the fine-tuning (FT) method, focusing on the effect of initial parameters. We perform one pass through the entire dataset to simulate the complete fine-tuning process. Consider the Mean Squared Error (MSE) loss function with parameters $V$ and $B$, where $B$ is represented as follows:

$$B = \begin{bmatrix} b_1^T \\ \vdots \\ b_m^T \\ b_{m+1}^T \\ \vdots \\ b_{m+C}^T \end{bmatrix},$$

where $b_i^T \in \mathbb{R}^{1 \times d}$ denotes the $i$-th row of matrix $B$, and $m + C = k$.

To apply one step of gradient descent, we need to compute the gradient of the loss function with respect to $V$, $b_1$, $b_2$, ..., $b_{m+C}$, and then perform one update step.

Let $V_0 = \begin{bmatrix} V_{com}^{*}{}^T & \mathbf{0}^T \end{bmatrix}^T$. It follows that:

$$\mathbb{E}\left[ \frac{\partial L}{\partial V} \bigg|_{\substack{V = V_0 \\ B = B_*}} \right] = \mathbb{E}_{x \sim \mathcal{D}_1} \left[ (V_0^T B_* x - V_1^{*T} B_* x) x^T B_*^T \right]$$

$$= \mathbb{E}_{x \sim \mathcal{D}_1} \left[ ((V_0 - V_1^*)^T B_* x) x^T B_*^T \right]$$

$$= (V_0 - V_1^*)^T B_* \left( \mathbb{E}_{x \sim \mathcal{D}_1} \left[ xx^T \right] \right) B_*^T$$

$$= (V_0 - V_1^*)^T B_* \left( \mathbb{E}_{z \sim \mathcal{N}(0, I_d)} \left[ (e_1 + \epsilon z)(e_1 + \epsilon z)^T \right] \right) B_*^T$$

$$= (V_0 - V_1^*)^T B_* \left( e_1 e_1^T + \epsilon^2 I_d \right) B_*^T$$

$$= (V_0 - V_1^*)^T B_* \left( e_1 e_1^T \right) B_*^T + (V_0 - V_1^*)^T B_* \left( \epsilon^2 I_d \right) B_*^T$$

$$= (V_0 - V_1^*)^T B_* \left( e_1 e_1^T \right) B_*^T + \epsilon^2 (V_0 - V_1^*)^T \qquad (B_* \text{ has orthonormal rows})$$

$$= (V_0 - V_1^*)^T \left( (B_*)_{:,1} (B_*)_{:,1}^T \right) + \epsilon^2 (V_0 - V_1^*)^T \qquad ((B_*)_{:,1} \text{ is first column of } B_*)$$

$$= \left( -\lambda (B_*)_{m+1,1} \right) (B_*)_{:,1}^T + \epsilon^2 (V_0 - V_1^*)^T.$$

Let $B_* = \begin{bmatrix} b_1^{*T} \\ \vdots \\ b_m^{*\,T} \\ b_{m+1}^{*\,T} \\ \vdots \\ b_{m+C}^{*\,T} \end{bmatrix}$. Then, it can be shown that:

$$\mathbb{E} \left[ \frac{\partial L}{\partial b_j} \Big|_{\substack{V=V_0 \\ B=B_*}} \right] = \mathbb{E}_{x \sim \mathcal{D}_1} \left[ (V_0^T B_* x - V_1^{*T} B_* x)(V_0)_j x^T \right]$$

$$= \mathbb{E}_{x \sim \mathcal{D}_1} \left[ (V_0)_j \left( (V_0 - V_1^*)^T B_* x \right) x^T \right]$$

$$= (V_0)_j (V_0 - V_1^*)^T B_* \left( \mathbb{E}_{x \sim \mathcal{D}_1} \left[ x x^T \right] \right)$$

$$= (V_0)_j (V_0 - V_1^*)^T B_* \left( \mathbb{E}_{z \sim \mathcal{N}(0, I_d)} \left[ (e_1 + \epsilon z)(e_1 + \epsilon z)^T \right] \right)$$

$$= (V_0)_j (V_0 - V_1^*)^T B_* \left( e_1 e_1^T + \epsilon^2 I_d \right)$$

$$= (V_0)_j (V_0 - V_1^*)^T B_* \left( e_1 e_1^T \right) + (V_0)_j (V_0 - V_1^*)^T B_* \left( \epsilon^2 I_d \right)$$

$$= (V_0)_j (V_0 - V_1^*)^T B_* \left( e_1 e_1^T \right) + \epsilon^2 (V_0)_j (V_0 - V_1^*)^T B_*.$$

Here, $(V_0)_j$ is the $j$-th element of the vector $V_0$. For learning rate $\eta$, one step of gradient descent can be:

$$V_{FT} = V_0 - \eta \left( \frac{\partial L}{\partial V} \Big|_{\substack{V=V_0 \\ B=B_*}} \right)^T$$

$$b_j^{FT} = b_j^* - \eta \left( \frac{\partial L}{\partial b_j} \Big|_{\substack{V=V_0 \\ B_0=B_*}} \right)^T.$$

These two equations can be further refined as:

$$\mathbb{E}[V_{FT}] = \begin{bmatrix} V_{com}^{*\,T} & \mathbf{0}^T \end{bmatrix}^T - \eta \left( -\lambda (B_*)_{m+1,1} (B_*)_{:,1} + \epsilon^2 (V_0 - V_1^*) \right)$$

$$= \begin{bmatrix} V_{com}^* \\ 0 \\ 0 \\ \vdots \\ 0 \end{bmatrix} + \begin{bmatrix} \mathbf{0} \\ \eta \lambda \epsilon^2 \\ 0 \\ \vdots \\ 0 \end{bmatrix} + \eta \lambda (B_*)_{m+1,1} (B_*)_{:,1}$$

$$\mathbb{E}[b_j^{FT}] = b_j^* - \eta \left( (V_0)_j (V_0 - V_1^*)^T B_* \left( e_1 e_1^T \right) + \epsilon^2 (V_0)_j (V_0 - V_1^*)^T B_* \right)^T$$

$$= b_j^* + \begin{bmatrix} \eta\lambda(V_0)_j(B_*)_{m+1,1} \\ 0 \\ \vdots \\ 0 \end{bmatrix} + \eta\lambda(V_0)_j\epsilon^2 b_{m+1}^*.$$

Therefore, we have:

$$\mathbb{E}\big[B_{FT}\big] = \begin{bmatrix} b_1^{*T} + \eta\lambda(V_0)_1\epsilon^2 {b_{m+1}^*}^T + \eta\lambda(V_0)_1(B_*)_{m+1,1}e_1^T \\ \vdots \\ {b_m^*}^T + \eta\lambda(V_0)_m\epsilon^2 {b_{m+1}^*}^T + \eta\lambda(V_0)_m(B_*)_{m+1,1}e_1^T \\ {b_{m+1}^*}^T + \eta\lambda(V_0)_{m+1}\epsilon^2 {b_{m+1}^*}^T + \eta\lambda(V_0)_{m+1}(B_*)_{m+1,1}e_1^T \\ \vdots \\ {b_{m+C}^*}^T + \eta\lambda(V_0)_{m+C}\epsilon^2 {b_{m+1}^*}^T + \eta\lambda(V_0)_{m+C}(B_*)_{m+1,1}e_1^T \end{bmatrix}$$

$$= \begin{bmatrix} b_1^{*T} + \eta\lambda(V_0)_1\epsilon^2 {b_{m+1}^*}^T + \eta\lambda(V_0)_1(B_*)_{m+1,1}e_1^T \\ \vdots \\ {b_m^*}^T + \eta\lambda(V_0)_m\epsilon^2 {b_{m+1}^*}^T + \eta\lambda(V_0)_m(B_*)_{m+1,1}e_1^T \\ {b_{m+1}^*}^T \\ \vdots \\ {b_{m+C}^*}^T \end{bmatrix}.$$

It can be shown that:

$$\begin{aligned}
\mathcal{L}_G(V_{FT}, B_{FT}) &= \frac{1}{C}\sum_{i\in[C]}\mathbb{E}_{x\sim\mathcal{D}_i}\left[\frac{1}{2}(V_{FT}^T B_{FT}x - V_i^{*T}B_*x)^2\right] \\
&= \frac{1}{2C}\sum_{i\in[C]}\mathbb{E}_{x\sim\mathcal{D}_i}\left[(B_{FT}^T V_{FT} - B_*^T V_i^*)^T xx^T(B_{FT}^T V_{FT} - B_*^T V_i^*)\right] \\
&= \frac{1}{2C}\sum_{i\in[C]}(B_{FT}^T V_{FT} - B_*^T V_i^*)^T\left[\mathbb{E}_{x\sim\mathcal{D}_i}xx^T\right](B_{FT}^T V_{FT} - B_*^T V_i^*) \\
&= \frac{1}{2C}\sum_{i\in[C]}(B_{FT}^T V_{FT} - B_*^T V_i^*)^T\left(e_ie_i^T + \epsilon^2 I_d\right)(B_{FT}^T V_{FT} - B_*^T V_i^*) \\
&= \frac{1}{2C}\sum_{i\in[C]}(B_{FT}^T V_{FT} - B_*^T V_i^*)^T\left(e_ie_i^T\right)(B_{FT}^T V_{FT} - B_*^T V_i^*) \\
&\quad + \frac{1}{2C}\sum_{i\in[C]}(B_{FT}^T V_{FT} - B_*^T V_i^*)^T\left(\epsilon^2 I_d\right)(B_{FT}^T V_{FT} - B_*^T V_i^*) \\
&= \frac{1}{2C}\sum_{i\in[C]}(B_{FT}^T V_{FT} - B_*^T V_i^*)^T\left(e_ie_i^T\right)(B_{FT}^T V_{FT} - B_*^T V_i^*) \\
&\quad + \epsilon^2\frac{1}{2C}\sum_{i\in[C]}\left\|(B_{FT}^T V_{FT} - B_*^T V_i^*)\right\|_2^2.
\end{aligned} \tag{4}$$

We have:

$$B_*^T V_i^* = \sum_{j=1}^{m}(V_{com}^*)_j b_j^* + \lambda b_{m+i}^*$$

$$\mathbb{E}\left[B_{FT}^T V_{FT}\right] = \eta\lambda\epsilon^2\sigma^2 b_{m+1}^* + \sum_{j=m+1}^{m+C}\left(\eta\lambda(B_*)_{m+1,1}(B_*)_{j,1}\right)b_j^*$$

$$+ \sum_{j=1}^{m}\left((V_{com}^*)_j + \eta\lambda(B_*)_{m+1,1}(B_*)_{j,1}\right)\left(b_j^* + \eta\lambda(V_{com}^*)_j\epsilon^2\sigma^2 b_{m+1}^* + \eta\lambda(V_{com}^*)_j(B_*)_{m+1,1}e_1\right).$$

Therefore, we can obtain:

$$\mathbb{E}\left[B_{FT}^T V_{FT} - B_*^T V_i^*\right] = \sum_{j=1}^{m}(V_{com}^*)_j\left(b_j^* + \eta\lambda(V_{com}^*)_j\epsilon^2\sigma^2 b_{m+1}^* + \eta\lambda(V_{com}^*)_j(B_*)_{m+1,1}e_1\right)$$

$$+ \eta\lambda\epsilon^2\sigma^2 b_{m+1}^* - \lambda b_{m+i}^* + \sum_{j=m+1}^{m+C}\left(\eta\lambda(B_*)_{m+1,1}(B_*)_{j,1}\right)b_j^*$$

$$+ \sum_{j=1}^{m}\left(\eta\lambda(B_*)_{m+1,1}(B_*)_{j,1}\right)\left(b_j^* + \eta\lambda(V_{com}^*)_j\epsilon^2\sigma^2 b_{m+1}^* + \eta\lambda(V_{com}^*)_j(B_*)_{m+1,1}e_1\right). \quad (5)$$

From equation (3), we observe that $\mathcal{L}_G(V_{LPFT}, B_{LPFT})$ is a monotonically increasing function of $\lambda$ and as $\lambda$ approaches zero, $\mathcal{L}_G(V_{LPFT}, B_{LPFT})$ also converges to zero. In contrast, combining equations (4) and (5), we find that $\mathcal{L}_G(V_{FT}, B_{FT})$ does not converge to zero as $\lambda$ approaches zero due to the presence of constant terms independent of $\lambda$. Note that $\mathcal{L}_G(V_{FT}, B_{FT})$ is non-negative by definition. While the $\lambda$-independent terms in (5) may take negative values, substituting this expression into (4) guarantees that the resulting global loss remains non-negative and, importantly, does not vanish as $\lambda \to 0$. Therefore, it follows that there always exists a threshold $\lambda^*$ such that for all $\lambda \leq \lambda^*$:

$$\mathcal{L}_G(V_{LPFT}, B_{LPFT}) \leq \mathcal{L}_G(V_{FT}, B_{FT}).$$

$\square$

**Corollary E.2** (Local performance under covariate and concept shift)**.** *Under the assumptions of Theorem 4.5, for any fixed client $i \in [C]$, there exists a threshold $\lambda^*$ such that for all $\lambda \leq \lambda^*$,*

$$\mathcal{L}_L^{(i)}(V_{LPFT}, B_{LPFT}) \ \leq \ \mathcal{L}_L^{(i)}(V_{FT}, B_{FT}).$$

*Proof.* The proof follows the same argument as in Corollary E.1. From the derivation of Theorem 4.5, the global loss of FT can be written as an average over client-specific losses, each consisting of a $\lambda$-dependent term and constants independent of $\lambda$. Fixing a client $i$ and isolating its contribution yields the corresponding local losses for FT and LP-FT. Since the $\lambda$-independent terms do not affect the ordering for sufficiently small $\lambda$, the existence of a threshold $\lambda^*$ ensuring $\mathcal{L}_L^{(i)}(V_{LPFT}, B_{LPFT}) \leq \mathcal{L}_L^{(i)}(V_{FT}, B_{FT})$ follows directly from the global comparison. $\square$

# F   Empirical Performance of LP-FT and FT Under Theorem 4.5 Conditions

To give a better understanding of Theorem 4.5, we give a simple visualization of two randomly generated data-generating functions for different clients and compute the global loss of LP-FT and FT based on equations (3) and (4).

These examples illustrate, within the theoretical setting of Sec. 4.2, the behavior of the loss functions for LP-FT and FT with a randomly generated labeling function $y = V_i^{*T}B_*x$, a fixed learning rate $\eta$, noise parameter $\epsilon$, and a fixed number of clients $C$. To compute this, we generated 1000 random matrices $B_*$ and 1000 randomly chosen linear heads $V_i^*$ as ground-truth labeling functions, ensuring they adhere to the theoretical assumptions. Using equations (3) and (4), we calculated the average loss of LP-FT and FT across these random trials.

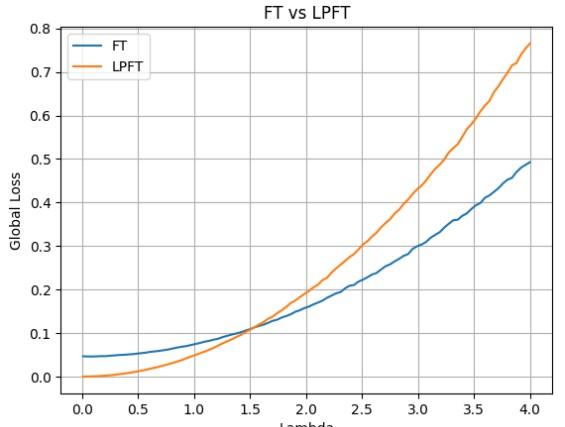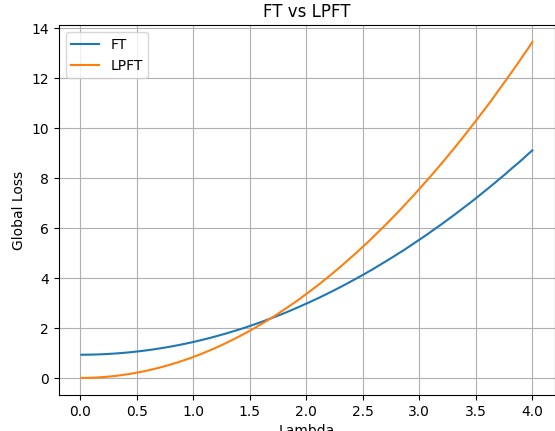

Figure 6: (a) Global loss of LP-FT and FT as a function of the heterogeneity parameter $\lambda$, with $\eta = 0.1$, $\epsilon = 0.1$, matrix $B_*$ as a $10 \times 20$ random matrix, and number of clients $C = 5$. (b) Global loss of LP-FT and FT as a function of the heterogeneity parameter $\lambda$, with $\eta = 0.1$, $\epsilon = 1$, matrix $B_*$ as a $10 \times 20$ random matrix, and number of clients $C = 5$.

As shown in Fig. 6, there exists a threshold $\lambda^*$ such that when $\lambda \leq \lambda^*$, LP-FT consistently outperforms FT. While this is a simplified example with a fixed number of clients, learning rate, noise parameter, and dimensionality of the ground-truth parameters $B_*$ and $V_i^*$, the observed trend remains similar across different parameter settings. The purpose of this figure is to provide an intuitive understanding of Theorem 4.5 in a controlled, simplified context. More comprehensive experiments in Sec. 5 demonstrate that LP-FT globally outperforms FT across a broader range of heterogeneity levels in real-world settings.

## G Computational Overhead of LP-FT Relative to FT

In this section, we study the computational cost of adding one step of linear probing (LP) before full fine-tuning (FT) to see how this additional LP step affects total computational cost. Suppose the dimension of the output of the feature extractor layer (i.e., the input to the linear head) is $d$, and the dimension of the output of the linear head is $m$. Thus, the linear head is a $d \times m$ linear layer. We assume there are $n$ samples. Throughout, we ignore the bias term, as its computational cost is negligible compared to that of the main matrix operations. Our goal is to determine the additional cost incurred by this LP stage compared to FT.

To estimate the computational cost of training a linear neural network layer with $d$ inputs, $m$ outputs, and $n$ samples, we analyze the steps involved:

**Forward pass** A linear neural network computes outputs as

$$Y = XW,$$

where $X \in \mathbb{R}^{n \times d}$ is the input matrix (with $n$ samples, each of dimension $d$), $W \in \mathbb{R}^{d \times l}$ is the weight matrix, and $Y \in \mathbb{R}^{n \times l}$ is the output matrix. The cost of this matrix multiplication is $O(ndl)$.

**Backward pass** To update $W$, the gradient of the loss $\mathcal{L}$ with respect to $W$ is

$$\nabla_W \mathcal{L} = X^\top (\nabla_Y \mathcal{L}).$$

Computing $\nabla_W \mathcal{L}$ involves: computing the gradient of the loss with respect to the outputs $Y$ (cost $O(nl)$), and performing the multiplication $X^\top (\nabla_Y \mathcal{L})$ (cost $O(ndl)$).

**Weight update**   If using gradient descent, the cost of updating the weights is $O(dl)$.

**Total Computational Cost:** The total cost for one forward and backward pass through the data is dominated by $O(ndl)$, which accounts for both forward propagation and gradient computation. If the training involves multiple epochs, the total cost scales as $O(e \cdot ndl)$, where $e$ is the number of epochs. During full fine-tuning, an additional backward pass through the feature extractor incurs a substantially larger cost $C_{\text{backbone}}$ per epoch. Therefore, adding an LP stage only increases the total cost by $O(e \cdot ndl)$, which is typically negligible compared to the cost of FT.

## Broader Impact

This paper presents work whose goal is to advance the field of Machine Learning. There are many potential societal consequences of our work, none of which we feel must be specifically highlighted here.

## Limitation

Our theoretical analysis focuses on concept shift and combined concept–covariate shift under simplified assumptions, such as linear models and a shared feature extractor. The analysis targets a specific failure mode in federated fine-tuning—*federated feature distortion*—which arises when local fine-tuning alters the shared representation due to heterogeneity in feature distributions and/or conditional label distributions given features. These settings correspond precisely to concept shift and combined concept–covariate shift, which are formalized by our theoretical results.

We do not provide a separate theoretical analysis of label shift, as it is not central to this feature-distortion mechanism. Under label shift, clients share the same class-conditional distribution $P_i(x \mid y) = P(x \mid y)$ and differ only in label priors $P_i(y)$, so heterogeneity stems from class imbalance rather than differences in learned features. For this reason, label shift does not directly instantiate the feature-distortion setting considered in our theory. Nevertheless, we include label-shift scenarios in our empirical evaluation, as they are widely studied in federated learning and of independent practical interest. Extensive empirical results across seven diverse datasets demonstrate that LP-FT consistently outperforms baseline methods in both personalization and generalization, even in complex, real-world scenarios. This suggests that the core insights behind LP-FT—such as mitigating feature distortion through a two-stage fine-tuning—extend beyond the theoretical scope and offer practical value for robust federated personalization.

