# OpenReview forum: "A Closer Look at Personalized Fine-Tuning in Heterogeneous Federated Learning"
_TMLR — Rejected by TMLR_

### Review · Reviewer_WNJW · 2025-12-09

**Summary Of Contributions:**

The main contribution of this paper lies in theoretically proving the effectiveness of the previous LP-FT (linear probing followed by full fine-tuning) strategy in alleviating the data heterogeneity challenge in federated learning. This paper also empirically uncovers that existing client-specific fine-tuning strategies disrupt the representation of global model and thus damage its global learning performance. This empirical observation is beneficial for designing new personalized fine-tune strategies. The proposed approach in this paper combines general federated learning algorithms and the LP-FT strategy, being an incremental technical contribution.

Strengths:

The proposed personalized fine-tune approach is simple and effective with provable theoretical advantages. The experimental results are also sufficient.

Weakness:

The theoretical results are restricted or need to be clarified.

(1) The results in Theorem 4.4 require strong conditions, such as linear independence of features, i.e., $\mathbb{E}[\mathbf{x}\mathbf{x}^\top]=\mathbf{I}$ and the conditions in Assumption 4.1, thereby making it highly restrictive.

(2) The correctness of Theorem 4.5 should be checked.

By the proof of Theorem 4.5, the constant term that is independent of $\lambda$ in Eq (5) should be positive. It is unclear to me whether it is positive or how to ensure that it is. To this end, it seems that we should carefully design $B_{\ast}$ and $V^{\ast}_{com}$. Besides, could the constant term be negative? If so, it seems that FT is better than LP-FT.

(3) There is a lack of theoretical analysis on the local performance on LP-FT.

This paper only proved that the global performance of LP-FT is better than FT, but did not establish similar results for the local performance.

(4) I also concern with the necessity of improving the global performance of each personalized model.

It is unclear to me why it is necessary to measure the global accuracy of each personalized model. In my opinion, we only need to measure the local accuracy of each personalized model restricted on its own test data. For some clients, if the distribution of data changes, then we just need to perform fine-tuning.

**Audience:**

Yes

**Audience Explanation:**

Federated learning is a prominent research direction in machine learning and artificial intelligence. Data heterogeneity is critical for both the theoretical research and application of federated learning. This paper empirically and theoretically shows a simple approach for alleviating the data heterogeneity. Thus I believe the TMLR community will be interested in knowing the findings of this paper.

**Broader Impact Concerns:**

I do not find any concerns on the ethical implications of this work.

**Claims And Evidence:**

Yes

**Claims Explanation:**

* In Appendix F, there are some simulated experiments that satisfy the conditions of Theorem 4.5. The experimental results verified the theoretical results.

* In Section 3.4 and Appendix D, the experimental results on benchmark datasets also verified the empirical performance of LP-FT.

**Requested Changes:**

* Please refer to weakness (2)-(4). These changes are critical to my final decision.

* There is also a minor suggestion. For the sake of clarity, it is better define the local accuracy and global accuracy by mathematical notations.

---

> ### Author Response · Authors · 2025-12-22
> **Response to Reviewer WNJW - Part 1**
>
> We thank the reviewer for their careful and thoughtful review and their detailed feedback on the theory, particularly Theorems 4.4 and 4.5. We address their questions and concerns below.
>
> **1. Theoretical Assumption (Assumption 4.1)**
> The results in Theorem 4.4 require strong conditions, such as linear independence of features, i.e. $\mathbb{E}[\mathbf{x}\mathbf{x}^\top] = \mathbf{I}$, and the conditions in Assumption 4.1, thereby making it highly restrictive.
>
> We thank the reviewer for highlighting this limitation. While we acknowledge that Assumption 4.1 and the isotropy assumption ($\mathbb{E}[\mathbf{x}\mathbf{x}^\top] = \mathbf{I}$) in Theorem 4.4 are strong, it is the standard protocol in theoretical machine learning to ensure mathematical tractability. This assumption is used to make the analysis of high-dimensional phenomena—such as optimization dynamics and generalization behavior—mathematically tractable. Indeed, **prominent related works rely on this exact setting to derive fundamental insights (e.g., Hastie et al., 2022; Ghorbani et al., 2019)**.
>
> Regarding Assumption 4.1, this modeling choice further serves to make the theoretical analysis tractable. Specifically, we introduce a two-layer data-generating process that enables a focused study of the interaction between pre-trained feature representations and linear prediction heads. To analyze our setting of PFT following the GFL stage, in Theorem 4.4, we assume that the pre-trained feature extractor recovers the true feature extractor, i.e., $B_0 = B_*$, while the pre-trained linear head captures only the common component, i.e., $V_0 = [V^*_{com}; 0]$. This assumption allows us to isolate and explain the effect of feature distortion introduced during the PFT stage and its impact on global performance.
>
> Notably, **the assumption $B_0 = B_*$ is consistent with Proposition 3.7 in Kumar et al. (2022)**, where it is similarly adopted to compare the performance of LP-FT and full fine-tuning. Ingeneral, this framework models clients’ heterogeneous data-generating functions through a shared feature extractor, capturing both shared structure and client-specific variability that is central to the PFT setting considered in this work.
>
> More broadly, while practical federated learning systems rely on deep non-linear models, it is standard for foundational theoretical analyses to adopt simplified architectures, such as deep linear or two-layer models, in order to obtain tractable insights under data heterogeneity (Charles & Konečný, 2021; Huang et al., 2021; Collins et al., 2022). In this spirit, our theoretical analysis complements our extensive empirical evaluations by elucidating the interplay between local updates and shared representations under personalization. We view this work as a principled first step toward understanding feature distortion in federated fine-tuning and as a foundation for future efforts to relax assumptions and extend the analysis to richer settings.
>
> We will include a dedicated discussion expanding on these points, should the paper be accepted.
>
> **References:**
> - Hastie, T., Montanari, A., Rosset, S., & Tibshirani, R. J. (2022). Surprises in high-dimensional ridgeless least squares interpolation. The Annals of Statistics, 50(2), 949-986.
> - Ghorbani, B., Mei, S., Misiakiewicz, T., & Montanari, A. (2019). Limitations of lazy training of two-layers neural network. Advances in Neural Information Processing Systems, 32.
> - Charles, Z., & Konečný, J. (2021). Convergence and accuracy trade-offs in federated learning and meta-learning. International Conference on Artificial Intelligence and Statistics (AISTATS).
> - Huang, B., Li, X., Zhao, P., & Huang, J. (2021). FL-NTK: A Neural Tangent Kernel-based Framework for Federated Learning Analysis. International Conference on Machine Learning (ICML).
> - Collins, L., Hassani, H., Mokhtari, A., & Shakkottai, S. (2022). FedAvg with fine-tuning: Local updates lead to representation learning. Advances in Neural Information Processing Systems (NeurIPS).
> - Kumar, A., Raghunathan, A., Jones, R., Ma, T., & Liang, P. (2022). Fine-tuning can distort pretrained features and underperform out-of-distribution. International Conference on Learning Representations (ICLR).

---

> > ### Author Response · Authors · 2025-12-22
> > **Response to Reviewer WNJW - Part 2**
> >
> > **2. Correctness of Theorem 4.5: While Eq. (5) might contain negative terms, its squared contribution to Eq. (4) ensures the FT global loss remains non-negative and non-vanishing, consistent with the result of Theorem 4.5.**
> >
> > We thank the reviewer for carefully checking Theorem 4.5 and raising this point. **Theorem 4.5 claims the existence of some $\lambda^* \geq 0$ such that for all $\lambda \le \lambda^*$ (corresponding to a low-heterogeneity regime)**, the global loss of LP-FT satisfies  $\mathcal{L}_{G}(V\_{LPFT}, B\_{LPFT}) \leq  \mathcal{L}\_{G}(V\_{FT}, B\_{FT})$.
> >
> >
> > We prove this by explicitly characterizing and comparing the global loss functions of LP-FT and FT. For LP-FT, in particular, Eq.(3) on page 28 demonstrates that $\mathcal{L}\_G(V\_{LPFT}, B\_{LPFT})$ is a monotonically increasing function of $\lambda$ and vanishes as $\lambda \to 0$. This behavior is further illustrated numerically in the low-heterogeneity regime of Figure 6.
> > For Full Fine-Tuning, Eq. (4) on page 30 defines the global loss as $\mathcal{L}\_G(V\_{FT}, B\_{FT}) = \frac{1}{2C}  \sum\limits_{i \in [C]}    (B\_{FT}^T V\_{FT} - B\_{*}^{T}V\_{i}^{\*})^{T} \Bigl(e_i e_i^T  \Bigr) (B\_{FT}^T V\_{FT} - B\_{\*}^{T}V\_{i}^{\*}) +  \epsilon^2   \frac{1}{2C} \sum\limits_{i \in [C]}  \left\\| (B\_{FT}^T V\_{FT} - B\_{\*}^{T}V\_{i}^{\*}) \right\\|_2^2$. By definition, this quantity is always non-negative, regardless of the model or ground-truth  parameters.
> >
> > The reviewer's concern stems from Eq. (5) (page 31), which characterizes $\mathbb{E}[B\_{FT}^T V\_{FT} - B\_{\*}^{T}V\_{i}^{\*}]$ and includes terms independent of $\lambda$. We emphasize that Eq. (5) does not represent the global loss $B\_{FT}^T V\_{FT} - B\_{\*}^{T}V\_{i}^{\*}$ itself; rather, it represents the expectation of the term $B_{FT}^T V_{FT} - B_*^T V_i^*$ appearing within the global loss formulation in Eq. (4). Although the $\lambda$-independent components in Eq. (5) may be negative, substituting this expression into Eq. (4) ensures that $B\_{FT}^T V\_{FT} - B\_{\*}^{T}V\_{i}^{\*}$ remains non-negative and, crucially, does not vanish as $\lambda \to 0$.
> >
> > Therefore, **the constant terms independent of $\lambda$ do not invalidate Theorem 4.5**, nor do they imply that FT outperforms LP-FT in the low-heterogeneity regime. Instead, **they explain why the global loss of FT remains bounded away from zero as $\lambda \to 0$**, whereas the global loss of LP-FT converges to zero, establishing the claimed performance separation. This theoretical behavior is numerically represented in the low-heterogeneity regime shown in Figure 6.

---

> ### Author Response · Authors · 2025-12-22
> **Response to Reviewer WNJW - Part 3**
>
> **3. Local Performance of LP-FT**
> There is a lack of theoretical analysis on the local performance on LP-FT. This paper only proved that the global performance of LP-FT is better than FT, but did not establish similar results for the local performance.
>
> We thank the reviewer for pointing out this observation regarding **local performance**. While our theoretical analysis in **Section 4** is presented in terms of **global performance**, the analysis is framed at the global level to highlight how **LP-FT mitigates federated feature distortion and preserves shared representations under heterogeneity**. Importantly, since the **global loss is defined as the average of client-specific local losses** (see page 9), the analysis in **Theorems 4.4 and 4.5 readily extends to local performance** by considering the loss of an individual client instead of the average across clients. Consequently, the same arguments apply when fixing a client $i$ and analyzing its local objective, leading to **analogous conclusions for local performance**.
>
> To illustrate this, consider **Theorem 4.4**. Following the same proof, **Eq. (2) on page 25** gives the global loss for full fine-tuning as
> $\mathcal{L}\_G(V\_{FT}, B\_{FT}) = \frac{\lambda^2}{2C} \bigl[ (\eta + \eta \sum\_{j=1}^m (V^{*}\_{com})\_j^2 - 1)^2 +  (C-1)((\eta + \eta \sum\_{j=1}^{m}(V^{\*}\_{com})_j^2)^2 + 1) \bigr]$.
>
> Extracting the corresponding **local loss for a fixed client** yields  $\mathcal{L}\_L(V\_{FT}, B\_{FT}) = \frac{\lambda^2}{2} (\eta + \eta \sum\_{j=1}^m (V^{\*}_{com})\_j^2 - 1)^2.$
>
> Similarly, for **LP-FT**, the analysis on **page 26** gives  $\mathcal{L}\_G(V\_{LPFT}, B\_{LPFT}) = \frac{\lambda^2}{2C} \bigl[(A\_{\alpha} - 1)^2 + (C-1)A\_{\alpha}^2\bigr],$ from which the **local loss** is  $\mathcal{L}\_L(V\_{LPFT}, B\_{LPFT}) = \frac{\lambda^2}{2} (A\_{\alpha} - 1)^2, $ which **converges to zero as $\alpha \to 1$**.
>
> Since $\lambda \ge 0$ and $(\eta + \eta \sum\_{j=1}^{m} (V^{\*}\_{com})\_{j}^{2} - 1)^2 \ge 0$, the **same ordering established in Theorem 4.4 immediately holds at the local level**. In particular, under the same conditions, we have $\mathcal{L}\_L(V\_{LPFT}, B\_{LPFT}) \le \mathcal{L}\_L(V\_{FT}, B\_{FT})$.
>
> An **analogous argument applies to Theorem 4.5**. For clarity and conciseness, we did not include a separate local-performance theorem in the main text, as the result follows directly from the global analysis *mutatis mutandis*. **For completeness, we will include the corresponding local versions of Theorems 4.4 and 4.5 in the final version of the paper**, should it be accepted.
>
> ---
>
> **4.Local and Global Accuracy**
>
> We thank the reviewer for this helpful suggestion. **To improve clarity, we will explicitly define local accuracy and global accuracy using precise mathematical notation** and include these definitions alongside the existing verbal descriptions to clearly specify the experimental setup. Let $\mathcal{D}\_i^{\mathrm{test}}$ denote the test set of client $i$ and $f\_{V\_i,B\_i}$ the personalized model after PFT for client $i$. The **local accuracy** is defined as $\mathrm{Local} = \frac{1}{C}\sum\_{i=1}^{C} \frac{1}{|\mathcal{D}\_i^{\mathrm{test}}|} \sum\limits_{(x,y)\in\mathcal{D}\_i^{\mathrm{test}}}\mathbf{1}\\{f\_{V\_i,B\_i}(x)=y\\},$ while the **global accuracy** is defined as $\mathrm{Global} = \frac{1}{C(C-1)}\sum\_{i=1}^C\sum\_{j\neq i}\frac{1}{|\mathcal{D}\_j^{\mathrm{test}}|}\sum\limits\_{(x,y)\in\mathcal{D}\_j^{\mathrm{test}}}\mathbf{1}\\{f\_{V\_i,B\_i}(x)=y\\}.$
> We will add this description in our revised version to highlight the metric of local and global accuracy.

---

> > ### Comment · Reviewer_WNJW · 2025-12-26
> >
> > I thank the authors for detailed rebuttal that have addressed most of my concerns.
> > For weakness (1)-(3), I agree with the authors' rebuttle.
> > However, the authors may omit addressing my last weakness.
> >
> > (4) I also concern with the necessity of improving the global performance of each personalized model.
> >
> > It is unclear to me why it is necessary to measure the global accuracy of each personalized model. In my opinion, we only need to measure the local accuracy of each personalized model restricted on its own test data. For some clients, if the distribution of data changes, then we just need to perform fine-tuning.

---

> > > ### Author Response · Authors · 2025-12-27
> > >
> > > We thank the reviewer for the opportunity to clarify our evaluation setup. The necessity of global performance evaluation is three-fold:
> > >
> > > **(a) Generalization in critical applications with data scarcity**. As stated in **our Introduction (Paragraph 3)**, PFT often causes models to overfit on local data. In critical applications with limited local data, such as disease diagnosis across multiple hospitals, a local model must not only perform well on current hospital but also generalize effectively to diverse patient populations (Xu et al., 2021). Therefore, balancing individual performance with global generalization is crucial in real-world reliability (Wu et al., 2022; Huang et al., 2024).
> > >
> > > **(b) Differentiating Positioning from LP-FT (Feature Distortion)**. Global performance is the key metric that differentiates our positioning from traditional pre-training and fine-tuning paradigms.
> > > * In the standard fine-tuning, typically only final local performance is considered.
> > > * However, Kumar et al. (2022) found that direct fine-tuning distorts pre-trained features, verifying this by showing a drop in Out-of-Distribution (OOD) performance.
> > > * In our setup, we directly monitor the **federated feature distortion (see Introduction Paragraph 7, Sec. 3.6)** by observing the drop in global performance. Maintaining high global accuracy provides a transparent guarantee that the model has retained a high-quality, generalizable feature representation rather than simply "memorizing" local data.
> > >
> > > **(c) Preserving the Core Value of Collaborative Federated Training**. From an optimization perspective, **ignoring global performance** allows local models to diverge so significantly in the weight space that **the "Federated" aspect of the learning ceases to provide value**. Monitoring global performance ensures that the personalized models still benefit from the features learned through global collaborative training, preventing them from reverting to isolated learners.

---

### Review · Reviewer_2LLN · 2025-12-09

**Summary Of Contributions:**

This paper investigates Personalized Fine-Tuning (PFT), a post-hoc and plug-and-play approach for achieving personalization in Federated Learning. The core contribution is the adaptation and in-depth emipirical analysis of LP-FT) a phased optimization strategy, to the FL setting. The authors identify federated feature distortion as the key mechanism driving the overfitting of standard PFT baselines and theoretically and empirically demonstrate that LP-FT effectively mitigates this issue. Through comprehensive experiments across seven datasets and diverse distribution shifts, LP-FT is established as a robust new baseline that effectively balances local personalization and global generalization.

Overall, the manuscript is methodologically sound, and the experimental results highly impactful. Additional, this paper is well-written, with visually appealing figures and tables, and the presentation of the key findings is very clear. The authors conducted experimental validation on a series of image datasets and provided theoretical results that corroborate the experimental outcomes. The overall logical coherence of the paper is exceptionally strong.

**Audience:**

Yes

**Audience Explanation:**

As the authors stated in the paper, PFT is a critical, lightweight alternative to process-integrated PFL methods, especially for real-world scenarios constrained by infrastructure lock-in or legacy FL protocols. This focus on a plug-and-play solution makes the work immediately relevant to the TMLR community.

**Broader Impact Concerns:**

This work presents a  emipirical study of federated learning, focusing on optimizing personalized fine-tuning strategies. I believe that it raises no significant ethical concerns.

**Claims And Evidence:**

Yes

**Claims Explanation:**

1. The study of this paper is highly systematic, benchmarking LP-FT against several other PFT variants across seven diverse datasets covering input shift, feature shift, and concept shift. The five evaluation metrics effectively capture the trade-off.

2. In addition to the comparative results, the studied concept of federated feature distortion, where local fine-tuning destabilizes globally learned features, is a critical mechanistic contribution. This provides a very clear explanation for the failure of standard fine-tuning in non-IID settings.

3. The authors provided theoretical results that corroborate the experimental outcomes.

**Requested Changes:**

1. The Introduction Section argues for PFT's practical necessity over process-integrated PFL methods. However, the results only compare between PFT variants. To truly assess the value of the PFT paradigm, should the comparison against a leading  process-integrated PFL methods be added?

2. The article shows that the proposed Linear Probing followed by Fine-Tuning (LP-FT) strategy performs well, and its computational cost is negligible compared to standard Fine-Tuning (FT). However, are there certain scenarios, such as in Out-of-Distribution (OOD) settings, where the LP approach may not work effectively, making full fine-tuning necessary?

3. Furthermore, fine-tuning strategies similar to LoRA have been widely validated as effective in many studies and experiments on large models. Could the analysis be expanded to include more discussion around LoRA? For instance, could experiments be added to compare fine-tuning only specific layers using LoRA?

---

> ### Author Response · Authors · 2025-12-22
> **Response to Reviewer 2LLN - Part 1**
>
> We thank the reviewer for their thoughtful review and positive assessment of the paper’s methodology, experiments, and clarity. We address their comments in detail below.
>
> **1. Comparison to process-integrated PFL methods**
> Within this PFT scope, we include a strong and up-to-date set of baselines covering the dominant post-hoc paradigms: Full FT, Proximal FT (FedProx-style regularization), Sparse FT, Model-Soup FT, and LSS FT (NeurIPS 2024)—a recent method explicitly proposed for FL personalization via local interpolation. This spectrum spans regularization-, sparsification-, and interpolation-based strategies commonly used in modern PFT.
>
> **We have included the PEFT with ViT (a non-CNN architecture) comparison in Table 8 in Appendix**. To further address the reviewer’s request for broader comparisons, we additionally benchmark FedAvg + LP-FT against multiple process-integrated PFL methods on CelebA under identical metrics. LP-FT achieves the best local, global, and average accuracy among the compared approaches, indicating that—even without modifying global training—LP-FT is competitive against substantially more complex process-integrated pipelines.
>
> **Comparison with Process-Integrated PFL Methods (CelebA)**
>
> | PFL Method | Local Acc. (↑) | Global Acc. (↑) | Avg. Acc. (↑) |
> |-----------|----------------|-----------------|---------------|
> | FedBN (Li et al., 2021) [1] | 88.68 | 61.35 | 61.00 |
> | PerAvg (Fallah et al., 2020) [2] | 87.06 | 67.26 | 66.66 |
> | FedNova (Wang et al., 2020) [3] | 88.68 | 54.26 | 53.41 |
> | FedRep (Collins et al., 2021) [4] | 86.94 | 52.99 | 52.57 |
> | FedSoup (Chen et al., 2023) [5] | 90.30 | 75.62 | 75.21 |
> | pFedFDA (McLaughlin et al., 2024) [6] | 90.41 | 76.56 | 76.63 |
> | FedL2G (Zhang et al., 2024) [7] | 92.46 | 79.27 | 78.82 |
> | **LP-FT (ours)** | **93.03** | **82.46** | **82.17** |
>
> **References:**
>
> [1] Li, X., Jiang, M., Zhang, X., Kamp, M., & Dou, Q. (2021). Fedbn: Federated learning on non-iid features via local batch normalization. arXiv preprint arXiv:2102.07623.
>
> [2] Fallah, A., Mokhtari, A., & Ozdaglar, A. (2020). Personalized federated learning: A meta-learning approach. arXiv preprint arXiv:2002.07948.
>
> [3] Wang, J., Liu, Q., Liang, H., Joshi, G., & Poor, H. V. (2020). Tackling the objective inconsistency problem in heterogeneous federated optimization. Advances in neural information processing systems, 33, 7611-7623.
>
> [4] Collins, L., Hassani, H., Mokhtari, A., & Shakkottai, S. (2021, July). Exploiting shared representations for personalized federated learning. In International conference on machine learning (pp. 2089-2099). PMLR.
>
> [5] Chen, M., Jiang, M., Dou, Q., Wang, Z., & Li, X. (2023, October). Fedsoup: Improving generalization and personalization in federated learning via selective model interpolation. In International Conference on Medical Image Computing and Computer-Assisted Intervention (pp. 318-328). Cham: Springer Nature Switzerland.
>
> [6] Mclaughlin, C., & Su, L. (2024). Personalized federated learning via feature distribution adaptation. Advances in Neural Information Processing Systems, 37, 77038-77059.
>
> [7] Zhang, J., Liu, Y., Hua, Y., Cao, J., & Yang, Q. (2024). Adaptive Guidance for Local Training in Heterogeneous Federated Learning. arXiv preprint arXiv:2410.06490.

---

> ### Author Response · Authors · 2025-12-22
> **Response to Reviewer 2LLN - Part 2**
>
> **2. Where LP-FT fails?**
> We thank the reviewer for this insightful question. **Yes, there are scenarios, particularly in severe out-of-distribution settings, where LP-FT may be less effective and full fine-tuning becomes necessary.** LP-FT is designed to preserve globally useful representations while allowing local adaptation, and is most effective when a reasonably pretrained feature extractor remains partially relevant across clients. **In our theoretical analysis, we explicitly assume $B_0 = B_*$**, which reflects the setting where clients share a common feature extraction mechanism.
>
> Importantly, **LP-FT does not remove the fine-tuning stage**. It performs fine-tuning after the linear probing step while keeping the feature extractor fixed. **The LP step allows adaptation at the prediction level while preventing unnecessary distortion of shared representations.** When distribution shifts primarily affect the prediction head, this controlled fine-tuning is often sufficient. **However, in more extreme OOD regimes where pretrained features are no longer appropriate and substantial feature relearning is required, full fine-tuning of the encoder may be necessary.**
>
> Finally, **LP-FT relies on the availability of a pretrained encoder and a clear separation between feature extraction and prediction layers**. In settings where such a pretrained encoder does not exist, or in architectures such as **large language models where representations and outputs are tightly coupled**, the direct applicability of LP-FT is limited and requires further investigation.
>
> ---
>
> **3. PEFT**
>
> We thank the reviewer for the thoughtful suggestion and would like to clarify that **LoRA-based results are already included in the paper**. Specifically, we report **transformer models with PEFT (including LoRA) in the appendix (Appendix C / Table 8)**, where **LP-FT consistently improves over standard fine-tuning under the same PFT setting**.
>
> We intentionally place these results in the appendix because **PEFT and LP-FT address orthogonal aspects of adaptation**. **PEFT methods such as LoRA focus on which parameters are trainable** to improve efficiency on large models, whereas **LP-FT focuses on how fine-tuning is staged over time to mitigate federated feature distortion**, which is the central mechanism studied in this paper. **Importantly, LP-FT is fully compatible with LoRA**, and the appendix results already demonstrate that **LP-FT’s benefits persist when combined with LoRA-style adaptations**.
>
> Expanding the main-text analysis to include **layer-wise LoRA variants or additional PEFT designs** would substantially shift the paper toward a **PEFT architecture study**, rather than a focused investigation of **post-hoc PFT mechanisms in federated learning**. For this reason, we keep PEFT results as **supporting evidence**, while centering the main text on insights that **generalize across architectures and parameterization choices**.
>
> We will revise the manuscript to more clearly highlight the **existing LoRA results and their implications**, and to explicitly state that **LP-FT complements—rather than competes with—LoRA and other PEFT methods**.

---

### Review · Reviewer_6AJx · 2025-12-10

**Summary Of Contributions:**

### Summary:

This paper proposes Linear Probing followed by Full Fine-Tuning (LP-FT), a straightforward extension of a previous centralized strategy to Federated Learning (FL). The motivation of the design is to balancing personalization and generalization in FL. LP-FT demonstrates effectiveness in alleviating the inherent shifts in FL and outperforms standard fine-tuning. The authors also provide theoretical justifications to better substantiate the superiority of their method.

### Pros:
[+]: The paper is well-written and easy to follow. I appreciate the authors' meticulous narrative.

[+]: The motivations are convincing, and the step-by-step exploration is commendable.

[+]: The proposed method is appreciated for its simplicity and focused design.

### Cons:
[-]: Limited novelty. As pointed out by the authors themselves, LP-FT only extends the previous centralized learning methods proposed by (a) to the FL setting. The authors' innovation is primarily in the empirical effectivenesss of a specific method to a new task, lacking a deeper exploration of more fundamental issues from a broader perspective. This renders LP-FT an incremental contribution rather than a novel one.

[-] The authors select covariate shift, concept shift, and label shift for analysis. While they provide theoretical underpinnings for the first two shifts, I found no theoretical analysis for label shift. Furthermore, the authors may have overlooked other shifts, such as prior shift, or the interplay among various types of shifts.

[-] The experimental backbone relies almost entirely on CNN-based models. Although a small experiment on ViT is provided in the appendix, this is far from sufficient to demonstrate the strong adaptability of LP-FT to different architectures. The authors should include experiments with a wider variety of ViT variants, as well as models beyond ViT (e.g., MLP-based models like Cyclemlp(b) or GNN-based models like ViG(c)). I also suggest moving these results to the main text rather than the appendix.

[-] The paper lacks experiments on large-scale datasets. Results obtained on several small- or medium-sized datasets cannot substantively demonstrate the method's superiority. The authors should incorporate large-scale datasets such as ImageNet, along with other challenging non-i.i.d. distributed datasets, to provide more substantial empirical evidence.

[-] The comparison methods are outdated. The authors have overlooked several mainstream federated learning methods, such as FedProx(d), FedBN(e), SWAD(f), FedLAW(g) and FedAWA(h). Furthermore, comparisons with Parameter-Efficient Fine-Tuning (PEFT) methods like LoRA and Adapter are only shown in the appendix. The selected PEFT methods are also primarily foundational early works rather than the latest established baselines in the field.

[-]: All theoretical analysis is built upon a two-layer MLP satisfying specific conditions. This severely limits the generalizability of the theory to real-world scenarios where more sophisticated architectures like ResNet and ViT are commonly employed.

[-]: The paper lacks ablation studies on general FL configurations, such as the number of clients, communication rounds, and client participation rate. This omission undermines the assessment of the method's robustness, learning efficiency, and deployment applicability.


### Refs:

(a) Fine-tuning can distort pretrained features and underperform out-of-distribution, ICLR 2022
(b) Cyclemlp: A mlp-like architecture for dense visual predictions, TPAMI 2023
(c) Vision gnn: An image is worth graph of nodes, NeurIPS 2022
(d) Federated optimization in heterogeneous networks, MLSys 2020
(e) FedBN: Federated learning on non-iid features via local batch normalization, ICLR 2021
(f) SWAD: domain generalization by seeking flat minima, NeurIPS 2021
(g) Revisiting weighted aggregation in federated learning with neural networks, ICML 2023
(h) FedAWA: Adaptive Optimization of Aggregation Weights in Federated Learning Using Client Vectors, CVPR 2025

**Audience:**

Yes

**Audience Explanation:**

I think that at least some researchers focused on Personalized Federated learning (PFL) would find the paper's findings relevant, as it offers a detailed analysis of extending LP-FT to post-hoc personalized fine-tuning and provides insights into federated feature distortion; however, its appeal is dampened by notable weaknesses, including incomplete theoretical coverage, insufficient experimental evidence, and the lack of ablation studies on key FL deployment configurations (e.g., number of clients, participation rate). These limitations may reduce interest for researchers seeking fully robust, real-world applicable solutions. Overall, the article may hold some appeal but likely does not meet the standards required for publication in TMLR.

**Broader Impact Concerns:**

None.

**Claims And Evidence:**

No

**Claims Explanation:**

The submission's core claims regarding LP-FT's superiority are supported by experiments and theoretical analyses for covariate and concept shifts. However, the overall convincingness is undermined by the lack of theoretical support for label shifts and a systematic exploration of mixed distribution shifts. More importantly, this paper provides insufficient empirical evidence from larger datasets and comparisons with stronger baselines. The absence of ablation studies on key practical FL deployment parameters also leaves critical gaps in evidence for real-world applicability.

**Requested Changes:**

[1]: Add theoretical analysis for label shift.
[2]: Incorporate analysis of other shifts (e.g., prior shift) and their combinations.
[3]: Add experimental results on larger-scale datasets.
[4]: Add comparisons with stronger, more recent baselines from mainstream FL families.
[5]: Conduct experiments with a broader range of backbones (including diverse ViT variants and other architectures like MLP/GNN-based models) and present these results in the main text.
[6]: Relax the theoretical assumptions of the framework to accommodate a wider variety of architectures beyond the two-layer MLP.
[7]: Provide ablation studies on general FL configurations.

---

> ### Author Response · Authors · 2025-12-22
> **Response to Reviewer 6AJx - Part 1**
>
> We appreciate the reviewer’s feedback; however, we respectfully disagree that our claims are undermined by the lack of empirical studies and theoretical analysis. We therefore first restate our targeted scope and setup, and then address each point in turn.
>
> **1. Novelty and Contribution**
>
> We would like to emphasize our novelty and contribution again here.
>
> **(a) Scope & Positioning**
>
> Our objective is **not to design another process-integrated PFL algorithm**, but to answer a **research question**:
> **once a standard global model is trained, how best to personalize it post-hoc to balance local and global performance?**
>
> To keep this question focused, we lock the global-training phase (**FedAvg**) and study only methods that act **after training**, with **no extra communication or server changes** (Figure 1 Caption). We would like to emphasize our novelty from the perspective of the methodology itself, the theory and the empirical validation.
>
> **(b) Methodological Contributions**
>
> Contrary to the perception that we simply transplant LP-FT into a federated setting, our work presents a **purposeful shift**: we apply LP-FT **exclusively during the final local training phase in FL** (Section 1 Paragraph 2, and Contribution 1).
>
> Unlike most personalized federated learning (PFL) methods that modify the entire training pipeline, our **plug-and-play approach** focuses on the **critical—but underexplored—last stage of local fine-tuning** (Table 1, Figure 1). This **simple yet effective design on FL** is a core strength, enabling broad deployability and improvements over existing techniques **without altering global training**.
>
>
> **(c) Theoretical Contributions**
>
> While centralized LP-FT assumes a **single ground-truth function**, our theoretical framework captures **multiple client-specific ground truths** (Section 1 Contribution 3).
>
> Specifically, we define data-generating functions with **unique linear heads for each client** and allow both **concept shifts and covariate shifts**—a stark contrast to centralized assumptions. These **FL-specific assumptions (Assumption 4.1)** are crucial for analyzing feature distortion under heterogeneous conditions.
>
> Furthermore, our **gradient-descent-based comparison of FT vs. LP-FT in FL** reveals why LP-FT preserves global features more effectively, offering **new insights into the interplay of local updates and shared representations**.
>
>
> **(d) Empirical Contributions**
>
> We extensively benchmark across **seven datasets and multiple distribution shifts** (Section 1 Contribution 2). Our findings expose **pronounced overfitting issues in standard PFT methods** (Figure 2), present **in-depth ablations** (Figure 3, Table 2), and connect our theory on heterogeneity (Table 4) and feature distortion to **real-world FL performance**.
>
> These experiments go beyond the scope of centralized LP-FT by exploring **more nuanced data shifts**, validating LP-FT’s robustness and shedding light on **why it outperforms existing personalization strategies**.
>
> Taken together, these points demonstrate that our method is **not a mere adaptation of LP-FT**, but a **tailored solution for FL personalization** that marries **simplicity, strong theory, and comprehensive empirical validation**.

---

> ### Author Response · Authors · 2025-12-22
> **Response to Reviewer 6AJx - Part 2**
>
> **2. Theoretical Concern on Two-Layer Linear Model**
>
> We believe this concern stems from **a misinterpretation of our theoretical analysis positioning**.
>
> **Our theoretical analysis follows the two-layer linear fine-tuning framework of Kumar et al. (2022) [1]**. As noted in Limitations, we adopt simplified assumptions, a two-layer data-generating process and a two-layer model (**Assumptions 4.1–4.2**), to isolate the interaction between a shared pre-trained feature extractor and linear heads. In particular, assuming recovery of the shared extractor ($B_0 = B_*$), consistent with **Proposition 3.7 in Kumar et al. [1]**, lets us cleanly study federated feature distortion caused by local fine-tuning under client heterogeneity, rather than conflating this effect with representation learning errors.
>
> **We emphasize that our goal is mechanistic insight, not an architecture-universal guarantee**: within this tractable setting, our gradient-descent analysis explains why Full FT can distort shared representations across clients while LP-FT better preserves globally useful features. This use of simplified models is standard in ML and FL theory to obtain tractable conclusions under heterogeneity (e.g., similar linear model setup also adopted in Charles & Konečný, 2021 [2]; Huang et al., 2021 [3]; Collins et al., 2022 [4]). We will add a short discussion in the revision to clarify our theoretical analysis positioning.
>
> **References:**
>
> [1] Kumar, A., Raghunathan, A., Jones, R., Ma, T., & Liang, P. (2022). Fine-tuning can distort pretrained features and underperform out-of-distribution. International Conference on Learning Representations (ICLR).
>
> [2] Charles, Z., & Konečný, J. (2021). Convergence and accuracy trade-offs in federated learning and meta-learning. International Conference on Artificial Intelligence and Statistics (AISTATS).
>
> [3] Huang, B., Li, X., Zhao, P., & Huang, J. (2021). FL-NTK: A Neural Tangent Kernel-based Framework for Federated Learning Analysis. International Conference on Machine Learning (ICML).
>
> [4] Collins, L., Hassani, H., Mokhtari, A., & Shakkottai, S. (2022). FedAvg with fine-tuning: Local updates lead to representation learning. Advances in Neural Information Processing Systems (NeurIPS).
>
> ---
>
> **3. Theory on Label Shift**
> **Our theory is not meant to cover all non-IID cases**. We include label shift empirically because it is a widely discussed non-IID setting in FL, and readers in this community may reasonably expect to see how LP-FT performs under it. Our theoretical analysis targets a specific failure mode in federated fine-tuning, **federated feature distortion**, where local fine-tuning changes the shared representation because clients differ in their **feature distribution and/or conditional label distribution given the features (i.e., covariate shift $P_i(x)$ and/or concept shift $P_i(y\mid x)$**; **Sec. 3.6, Sec. 4**). Therefore, these relevant concept shift and combined concept–covariate shift are formalized by our theorems.
>
> **Label shift is not central to this setup**. Under label shift, clients share the **same class-conditional distribution $P_i(x\mid y)=P(x\mid y)$** and **differ only in label priors $P_i(y)$**. Thus, the heterogeneity comes from class proportion imbalance rather than differences in features. For this reason, label shift does not directly instantiate the feature-distortion setting in our theory (as noted in **Sec. Limitations**).
>
>
> ---

---

> ### Author Response · Authors · 2025-12-22
> **Response to Reviewer 6AJx - Part 3**
>
> **4. Clarification on Our Setting and Baselines**
>
> We respectfully disagree that our empirical comparisons lack rigor. **We do not position LP-FT as a universal SOTA method spanning all process-integrated PFL and the full PEFT literature**; rather, the paper’s goal is to rigorously characterize a simple, post-hoc PFT approach (LP-FT) and show—through comprehensive experiments and supporting theory—that LP-FT can consistently balance local personalization and global generalization. Accordingly, our evaluation is designed to be thorough within the **post-hoc PFT setting**, and **our baselines are chosen to be strong, contemporary, and directly compatible with this deployment setup**. We elaborate on the positioning, breadth of datasets/shifts and the included baselines below.
>
> (a) Positioning and setting of the paper.
> **Our goal is not to introduce a new state-of-the-art process-integrated FL algorithm or to claim dominance over all personalized FL pipelines**. Rather, we **provide a comprehensive and rigorous analysis of a simple, post-hoc, plug-and-play personalization method (LP-FT in PFT)** showing that it can achieve a strong local–global trade-off across a wide range of non-IID conditions without redesigning the federated training protocol. We believe this is a valuable message for the FL community: a simple LP-FT in PFT can serve as a strong and generalizable baseline achieving good balanced local and global performance across a wide range of distribution shifts.
>
> Accordingly, **we keep the global federated optimization fixed (vanilla FedAvg with a standard participation schedule/round budget)** and study local and global performance to distribution shifts via post-hoc personalization, so **ablations over client participation rates or communication rounds are out of scope for our paper’s positioning**.
>
> (b) Our empirical study is broad under diverse distribution shifts, datasets, and models.
> We respectfully disagree that the experiments are narrow. We benchmark LP-FT extensively across **seven datasets and four distribution shifts** (see **Sec. 1, Contribution 2**). Specifically, we evaluate LP-FT across seven heterogeneous datasets spanning input-level corruption (CIFAR10-C, CIFAR100-C), feature-level domain shifts (Digit5, DomainNet), and spurious correlation–based concept shifts (CheXpert, CelebA), using up to 50 simulated clients. Beyond reporting performance, we (i) clearly show and analyze overfitting issues of standard PFT variants (**Fig. 2**), (ii) provide in-depth ablations and sensitivity analyses (**Fig. 3, Tab. 2**), and (iii) connect our theoretical heterogeneity/feature-distortion mechanism to practical FL behavior (**App. Tab. 6**). Importantly, these settings go beyond the centralized LP-FT analysis by considering federated-specific, nuanced shifts, thereby validating LP-FT’s robustness and clarifying why it outperforms common personalization strategies.

---

> > ### Author Response · Authors · 2025-12-22
> > **Response to Reviewer 6AJx - Part 4**
> >
> > (c) Baseline selection.
> > We emphasize that **our paper targets post-hoc personalized fine-tuning** (PFT): personalization applied after FedAvg global training, with no changes to the FL protocol or server–client coordination. Therefore, our primary comparators are intentionally restricted to plug-and-play PFT baselines that are compatible with this deployment setting. Process-integrated PFL methods (e.g., personalized aggregation, clustering, iterative server feedback) are not directly comparable because they fundamentally alter the global training pipeline and solve a different problem regime.
> >
> > Within this PFT scope, we include a strong and up-to-date set of baselines covering the dominant post-hoc paradigms: Full FT, Proximal FT (FedProx-style regularization), Sparse FT, Model-Soup FT, and LSS FT (NeurIPS 2024)—a recent method explicitly proposed for FL personalization via local interpolation. This spectrum spans regularization-, sparsification-, and interpolation-based strategies commonly used in modern PFT.
> >
> > **We have included the PEFT with ViT (a non-CNN architecture) comparison in Table 8 in Appendix**. To further address the reviewer’s request for broader comparisons, we additionally benchmark FedAvg + LP-FT against multiple process-integrated PFL methods on CelebA under identical metrics. LP-FT achieves the best local, global, and average accuracy among the compared approaches, indicating that—even without modifying global training—LP-FT is competitive against substantially more complex process-integrated pipelines.
> >
> > **Comparison with Process-Integrated PFL Methods (CelebA)**
> >
> > | PFL Method | Local Acc. (↑) | Global Acc. (↑) | Avg. Acc. (↑) |
> > |-----------|----------------|-----------------|---------------|
> > | FedBN (Li et al., 2021) [1] | 88.68 | 61.35 | 61.00 |
> > | PerAvg (Fallah et al., 2020) [2] | 87.06 | 67.26 | 66.66 |
> > | FedNova (Wang et al., 2020) [3] | 88.68 | 54.26 | 53.41 |
> > | FedRep (Collins et al., 2021) [4] | 86.94 | 52.99 | 52.57 |
> > | FedSoup (Chen et al., 2023) [5] | 90.30 | 75.62 | 75.21 |
> > | pFedFDA (McLaughlin et al., 2024) [6] | 90.41 | 76.56 | 76.63 |
> > | FedL2G (Zhang et al., 2024) [7] | 92.46 | 79.27 | 78.82 |
> > | **LP-FT (ours)** | **93.03** | **82.46** | **82.17** |
> >
> > **References:**
> >
> > [1] Li, X., Jiang, M., Zhang, X., Kamp, M., & Dou, Q. (2021). Fedbn: Federated learning on non-iid features via local batch normalization. arXiv preprint arXiv:2102.07623.
> >
> > [2] Fallah, A., Mokhtari, A., & Ozdaglar, A. (2020). Personalized federated learning: A meta-learning approach. arXiv preprint arXiv:2002.07948.
> >
> > [3] Wang, J., Liu, Q., Liang, H., Joshi, G., & Poor, H. V. (2020). Tackling the objective inconsistency problem in heterogeneous federated optimization. Advances in neural information processing systems, 33, 7611-7623.
> >
> > [4] Collins, L., Hassani, H., Mokhtari, A., & Shakkottai, S. (2021, July). Exploiting shared representations for personalized federated learning. In International conference on machine learning (pp. 2089-2099). PMLR.
> >
> > [5] Chen, M., Jiang, M., Dou, Q., Wang, Z., & Li, X. (2023, October). Fedsoup: Improving generalization and personalization in federated learning via selective model interpolation. In International Conference on Medical Image Computing and Computer-Assisted Intervention (pp. 318-328). Cham: Springer Nature Switzerland.
> >
> > [6] Mclaughlin, C., & Su, L. (2024). Personalized federated learning via feature distribution adaptation. Advances in Neural Information Processing Systems, 37, 77038-77059.
> >
> > [7] Zhang, J., Liu, Y., Hua, Y., Cao, J., & Yang, Q. (2024). Adaptive Guidance for Local Training in Heterogeneous Federated Learning. arXiv preprint arXiv:2410.06490.

---

### Review · Reviewer_jeK7 · 2025-12-10

**Summary Of Contributions:**

This paper investigates post-hoc Personalized Fine-Tuning (PFT), an easily deployable method for balancing personalized performance against global model retention in Federated Learning (FL). The authors adapt a centralized strategy, Linear Probing followed by Full Fine-Tuning (LP-FT), to the FL setting. The central conceptual contribution is the identification and empirical validation of federated feature distortion, arguing that standard fine-tuning destabilizes the feature extractor learned globally, which LP-FT's phased updates successfully mitigates. The work provides extensive empirical results across various distribution shifts and theoretical backing for LP-FT's global performance superiority over Full Fine-Tuning (FT) under idealized model assumptions.

***Strengths:***

- Mechanistic Insight: The discovery and quantification of federated feature distortion provide a valuable, actionable insight into the core failure mode of local fine-tuning in non-IID FL, serving as a clear research direction.

- Empirical Performance (on constrained scope): LP-FT consistently demonstrates better performance in balancing personalization (local accuracy) and generalization (global accuracy) compared to its chosen fine-tuning counterparts across the tested datasets.

- Simplicity and Deployability: PFT, and by extension LP-FT, is a crucial lightweight, post-hoc strategy that ensures broad compatibility and deployment robustness without necessitating a redesign of the global FL framework.

***Weaknesses:***

- Theoretical Fragility: The underpinning of the theoretical claims rests on a highly oversimplified model assumption (two-layer linear network). This structural constraint severely limits the external validity of the derived theorems regarding modern deep learning architectures (e.g., ResNets, Transformers).

- Narrow Experimental Scope: The empirical validation is focused on smaller-scale image datasets, lacking a rigorous demonstration of LP-FT's effectiveness and scalability on large-scale, high-dimensional datasets prevalent in current research.

- Absence of Comparative Rigor: The set of comparator methods is narrow and outdated, omitting contemporary and impactful personalized FL methods and recent advanced parameter-efficient fine-tuning (PEFT) techniques, thus not fully justifying LP-FT as the most robust modern baseline .


Incomplete Distribution Shift Coverage: A formal, theoretical justification for the method's behavior under label shift is conspicuously missing, despite its empirical inclusion.

**Audience:**

Yes

**Audience Explanation:**

The subject matter—addressing heterogeneity in FL via lightweight personalization—is highly relevant to the TMLR audience. The paper offers a valuable contribution by articulating the federated feature distortion phenomenon, providing a new diagnostic lens and a simple, implementable solution (LP-FT) for practitioners. This makes the findings of immediate interest to those focused on robust and scalable decentralized learning systems.

**Broader Impact Concerns:**

The work is focused on enhancing the performance of machine learning models in a privacy-preserving, decentralized setting. By providing a more robust personalization technique, the work contributes positively to the deployment of safer and more accurate models in sensitive domains like healthcare. No ethical concerns requiring further attention were identified.

**Claims And Evidence:**

No

**Claims Explanation:**

The overall claims are undermined by a lack of evidence demonstrating the scalability and generalizability of the proposed solution outside of a controlled, small-scale context.

- Generalizability Gap: The core theoretical justifications rely on a highly constrained architectural model (two-layer linear network, Assumption 4.2). While the authors present a limited empirical attempt using deep models, the theoretical framework does not credibly connect to the success of LP-FT in non-linear deep learning settings, creating a significant gap between the derived conclusions and the demonstrated practical performance.

- Insufficient Stress Testing: The empirical setting does not adequately reflect the complexity of modern FL deployments. The exclusive use of small- to medium-scale benchmarks and the reliance on a limited set of comparators do not furnish sufficient evidence to definitively establish LP-FT's superiority against a broader spectrum of advanced FL and fine-tuning strategies. Robustness to scale and architectural diversity is not convincingly demonstrated.

- Lacking Foundational Analysis: The theoretical coverage of non-IID challenges is incomplete, as the rationale and analytical outcome for the label shift scenario are not formalized, despite its inclusion in the empirical evaluation. This indicates a gap in the systematic theoretical investigation.

**Requested Changes:**

The following adjustments are necessary to address the identified limitations:

- Elevate Theoretical Foundation: The fundamental architectural assumption (two-layer linear network) must be significantly addressed. The authors need to provide a deeper analytical discussion explaining the mechanism by which the linear-model insights extrapolate to the deep, non-linear networks used in practice, or, alternatively, explore paths to relax the linearity assumption in the theoretical model.

- Complete Theoretical Coverage: A formal theoretical analysis for the label shift scenario must be included to ensure a comprehensive and complete theoretical foundation for the distribution shifts evaluated empirically.

- Expand Empirical Scope for Scalability: New comparative experiments must be conducted on at least one large-scale, modern FL benchmark using high-capacity backbones to demonstrate LP-FT's effectiveness and scalability in a realistic, non-constrained setting.

- Enhance Comparative Rigor: The empirical study must be expanded to include comparisons against a broader set of strong, contemporary baselines, including state-of-the-art methods in both process-integrated Personalized FL and modern Parameter-Efficient Fine-Tuning (PEFT) to justify the claim of being a robust baseline.

- Validate Architectural Agnosticism: The empirical evaluation on non-CNN architectures must be expanded and moved from the appendix to the main text, using a wider variety of modern backbones (e.g., Transformers) to convincingly prove LP-FT's general adaptability.

- Assess Deployment Robustness: Crucial ablation studies on practical FL configuration variables—such as the client participation rate and the impact of varying communication rounds—must be provided to validate the method's deployment robustness and efficiency in real-world environments.

---

> ### Author Response · Authors · 2025-12-22
> **Response to Reviewer jeK7 - Part 1**
>
> We appreciate the reviewer’s feedback, however, we respectfully disagree with several of the “must” requests, as they appear to stem from a misunderstanding of our paper’s positioning. We therefore first restate our targeted scope and setup, and then address each point in turn.
>
> **1. Theoretical Concern on Two-Layer Linear Model**
>
> We believe this concern stems from **a misinterpretation of our theoretical analysis positioning**.
>
> **Our theoretical analysis follows the two-layer linear fine-tuning framework of Kumar et al. (2022) [1]**. As noted in Limitations, we adopt simplified assumptions, a two-layer data-generating process and a two-layer model (**Assumptions 4.1–4.2**), to isolate the interaction between a shared pre-trained feature extractor and linear heads. In particular, assuming recovery of the shared extractor ($B_0 = B_*$), consistent with **Proposition 3.7 in Kumar et al. [1]**, lets us cleanly study federated feature distortion caused by local fine-tuning under client heterogeneity, rather than conflating this effect with representation learning errors.
>
> **We emphasize that our goal is mechanistic insight, not an architecture-universal guarantee**: within this tractable setting, our gradient-descent analysis explains why Full FT can distort shared representations across clients while LP-FT better preserves globally useful features. This use of simplified models is standard in ML and FL theory to obtain tractable conclusions under heterogeneity (e.g., similar simplified model setup also adopted in Charles & Konečný, 2021 [2]; Huang et al., 2021 [3]; Collins et al., 2022 [4]). We will add a short discussion in the revision to clarify our theoretical analysis positioning.
>
> [1] Kumar, A., Raghunathan, A., Jones, R., Ma, T., & Liang, P. (2022). Fine-tuning can distort pretrained features and underperform out-of-distribution. International Conference on Learning Representations (ICLR).
> [2] Charles, Z., & Konečný, J. (2021). Convergence and accuracy trade-offs in federated learning and meta-learning. International Conference on Artificial Intelligence and Statistics (AISTATS).
> [3] Huang, B., Li, X., Zhao, P., & Huang, J. (2021). FL-NTK: A Neural Tangent Kernel-based Framework for Federated Learning Analysis. International Conference on Machine Learning (ICML).
> [4] Collins, L., Hassani, H., Mokhtari, A., & Shakkottai, S. (2022). FedAvg with fine-tuning: Local updates lead to representation learning. Advances in Neural Information Processing Systems (NeurIPS).
>
> ---
>
> **2. Theory on Label Shift**
>
> **Our theory is not meant to cover all non-IID cases**. We include label shift empirically because it is a widely discussed non-IID setting in FL, and readers in this community may reasonably expect to see how LP-FT performs under it. Our theoretical analysis targets a specific failure mode in federated fine-tuning, **federated feature distortion**, where local fine-tuning changes the shared representation because clients differ in their **feature distribution and/or conditional label distribution given the features (i.e., covariate shift $P_i(x)$ and/or concept shift $P_i(y\mid x)$**; **Sec. 3.6, Sec. 4**). Therefore, these relevant concept shift and combined concept–covariate shift are formalized by our theorems.
>
> **Label shift is not central to this setup**. Under label shift, clients share the **same class-conditional distribution $P_i(x\mid y)=P(x\mid y)$** and **differ only in label priors $P_i(y)$**. Thus, the heterogeneity comes from class proportion imbalance rather than differences in features. For this reason, label shift does not directly instantiate the feature-distortion setting in our theory (as noted in **Sec. Limitations**).
>
> ---

---

> > ### Author Response · Authors · 2025-12-22
> > **Response to Reviewer jeK7 - Part 2**
> >
> > **3. Clarification on Our Setting and Baselines**
> >
> > We respectfully disagree that our empirical comparisons lack rigor. **We do not position LP-FT as a universal SOTA method spanning all process-integrated PFL and the full PEFT literature**; rather, the paper’s goal is to rigorously characterize a simple, post-hoc PFT approach (LP-FT) and show—through comprehensive experiments and supporting theory—that LP-FT can consistently balance local personalization and global generalization. Accordingly, our evaluation is designed to be thorough within the **post-hoc PFT setting**, and **our baselines are chosen to be strong, contemporary, and directly compatible with this deployment setup**. We elaborate on the positioning, breadth of datasets/shifts and the included baselines below.
> >
> > (1) Positioning and setting of the paper.
> > **Our goal is not to introduce a new state-of-the-art process-integrated FL algorithm or to claim dominance over all personalized FL pipelines**. Rather, we **provide a comprehensive and rigorous analysis of a simple, post-hoc, plug-and-play personalization method (LP-FT in PFT)** showing that it can achieve a strong local–global trade-off across a wide range of non-IID conditions without redesigning the federated training protocol. We believe this is a valuable message for the FL community: a simple LP-FT in PFT can serve as a strong and generalizable baseline achieving good balanced local and global performance across a wide range of distribution shifts.
> >
> > Accordingly, **we keep the global federated optimization fixed (vanilla FedAvg with a standard participation schedule/round budget)** and study local and global performance to distribution shifts via post-hoc personalization, so **ablations over client participation rates or communication rounds are out of scope for our paper’s positioning**.
> >
> > (2) Our empirical study is broad under diverse distribution shifts, datasets, and models.
> > We respectfully disagree that the experiments are narrow. We benchmark LP-FT extensively across **seven datasets and four distribution shifts** (see **Sec. 1, Contribution 2**). Specifically, we evaluate LP-FT across seven heterogeneous datasets spanning input-level corruption (CIFAR10-C, CIFAR100-C), feature-level domain shifts (Digit5, DomainNet), and spurious correlation–based concept shifts (CheXpert, CelebA), using up to 50 simulated clients. Beyond reporting performance, we (i) clealy show and analyze overfitting issues of standard PFT variants (**Fig. 2**), (ii) provide in-depth ablations and sensitivity analyses (**Fig. 3, Tab. 2**), and (iii) connect our theoretical heterogeneity/feature-distortion mechanism to practical FL behavior (**App. Tab. 6**). Importantly, these settings go beyond the centralized LP-FT analysis by considering federated-specific, nuanced shifts, thereby validating LP-FT’s robustness and clarifying why it outperforms common personalization strategies.

---

> > > ### Author Response · Authors · 2025-12-22
> > > **Response to Reviewer jeK7 - Part 3**
> > >
> > > (3) Baseline selection.
> > > We emphasize that **our paper targets post-hoc personalized fine-tuning** (PFT): personalization applied after FedAvg global training, with no changes to the FL protocol or server–client coordination. Therefore, our primary comparators are intentionally restricted to plug-and-play PFT baselines that are compatible with this deployment setting. Process-integrated PFL methods (e.g., personalized aggregation, clustering, iterative server feedback) are not directly comparable because they fundamentally alter the global training pipeline and solve a different problem regime.
> > >
> > > Within this PFT scope, we include a strong and up-to-date set of baselines covering the dominant post-hoc paradigms: Full FT, Proximal FT (FedProx-style regularization), Sparse FT, Model-Soup FT, and LSS FT (NeurIPS 2024)—a recent method explicitly proposed for FL personalization via local interpolation. This spectrum spans regularization-, sparsification-, and interpolation-based strategies commonly used in modern PFT.
> > >
> > > **We have included the PEFT with ViT (a non-CNN architecture) comparison in Table 8 in Appendix**. To further address the reviewer’s request for broader comparisons, we additionally benchmark FedAvg + LP-FT against multiple process-integrated PFL methods on CelebA under identical metrics. LP-FT achieves the best local, global, and average accuracy among the compared approaches, indicating that—even without modifying global training—LP-FT is competitive against substantially more complex process-integrated pipelines.
> > >
> > > **Comparison with Process-Integrated PFL Methods (CelebA)**
> > >
> > > | PFL Method | Local Acc. (↑) | Global Acc. (↑) | Avg. Acc. (↑) |
> > > |-----------|----------------|-----------------|---------------|
> > > | FedBN (Li et al., 2021) [1] | 88.68 | 61.35 | 61.00 |
> > > | PerAvg (Fallah et al., 2020) [2] | 87.06 | 67.26 | 66.66 |
> > > | FedNova (Wang et al., 2020) [3] | 88.68 | 54.26 | 53.41 |
> > > | FedRep (Collins et al., 2021) [4] | 86.94 | 52.99 | 52.57 |
> > > | FedSoup (Chen et al., 2023) [5] | 90.30 | 75.62 | 75.21 |
> > > | pFedFDA (McLaughlin et al., 2024) [6] | 90.41 | 76.56 | 76.63 |
> > > | FedL2G (Zhang et al., 2024) [7] | 92.46 | 79.27 | 78.82 |
> > > | **LP-FT (ours)** | **93.03** | **82.46** | **82.17** |
> > >
> > > **References:**
> > >
> > > [1] Li, X., Jiang, M., Zhang, X., Kamp, M., & Dou, Q. (2021). Fedbn: Federated learning on non-iid features via local batch normalization. arXiv preprint arXiv:2102.07623.
> > >
> > > [2] Fallah, A., Mokhtari, A., & Ozdaglar, A. (2020). Personalized federated learning: A meta-learning approach. arXiv preprint arXiv:2002.07948.
> > >
> > > [3] Wang, J., Liu, Q., Liang, H., Joshi, G., & Poor, H. V. (2020). Tackling the objective inconsistency problem in heterogeneous federated optimization. Advances in neural information processing systems, 33, 7611-7623.
> > >
> > > [4] Collins, L., Hassani, H., Mokhtari, A., & Shakkottai, S. (2021, July). Exploiting shared representations for personalized federated learning. In International conference on machine learning (pp. 2089-2099). PMLR.
> > >
> > > [5] Chen, M., Jiang, M., Dou, Q., Wang, Z., & Li, X. (2023, October). Fedsoup: Improving generalization and personalization in federated learning via selective model interpolation. In International Conference on Medical Image Computing and Computer-Assisted Intervention (pp. 318-328). Cham: Springer Nature Switzerland.
> > >
> > > [6] Mclaughlin, C., & Su, L. (2024). Personalized federated learning via feature distribution adaptation. Advances in Neural Information Processing Systems, 37, 77038-77059.
> > >
> > > [7] Zhang, J., Liu, Y., Hua, Y., Cao, J., & Yang, Q. (2024). Adaptive Guidance for Local Training in Heterogeneous Federated Learning. arXiv preprint arXiv:2410.06490.

---

### Author Response · Authors · 2026-01-09
**General Response**

We thank the reviewers for their thoughtful evaluation and insightful comments and suggestions.

We appreciate the reviewers’ acknowledgment that this work advances both understanding and practice in federated personalization. Specifically, reviewers highlighted that our work provides **'valuable, actionable insight'** (Reviewer jeK7) into the **'key mechanism' of feature distortion** (Reviewer 2LLN) through a **'convincing' and 'commendable step-by-step exploration'** (Reviewer 6AJx). Furthermore, they noted that LP-FT serves as a **'robust'** (Reviewer 2LLN) and **'effective'** (Reviewer WNJW) baseline that is highly practical due to its **'simplicity and focused design'** (Reviewer 6AJx) and **'deployment robustness'** (Reviewer jeK7).

During the rebuttal, we made **a focused set of targeted updates and clarifications to improve clarity, positioning, and completeness**, while keeping the paper’s core contributions unchanged. The updates primarily (i) **clarify the paper’s scope and intended claims**, especially the distinction between post-hoc PFT and process-integrated PFL, and our positioning of LP-FT as a plug-and-play baseline; (ii) **add additional empirical comparisons** against representative PFL and PEFT baselines **to better contextualize LP-FT**, and (iii) **refine the theoretical presentation** by clarifying assumptions, providing local counterparts to key results, expanding the discussion of label shift, formalizing evaluation metrics, and adding proof-level clarifications.

---

> ### Author Response · Authors · 2026-01-09
> **Summary of Updates**
>
> Following the rebuttal and the reviewers’ feedback, we have incorporated revisions into the manuscript. Below, we summarize the changes made in the revised version.
>
> **1. Clarification of Scope: PFT vs. Process-Integrated PFL.**
> We added clarifying explanations in the Introduction distinguishing *process-integrated personalized federated learning (PFL)* methods from *post-hoc personalized fine-tuning (PFT)*, which is the primary focus of this paper. We explicitly state that our goal is not to propose a new process-integrated PFL algorithm, but rather to rigorously characterize **plug-and-play, post-hoc personalization applied after a fixed global training stage**, and to establish LP-FT as a strong and practical baseline in this setting.
>
> **2. Additional Comparisons with Process-Integrated PFL Methods.**
> To further contextualize the effectiveness of post-hoc PFT, we added new experiments comparing **FedAvg + LP-FT** against representative process-integrated PFL methods on the **CelebA** dataset (see **Table 8**), including FedBN, PerAvg, FedNova, FedRep, FedSoup, pFedFDA, and FedL2G.
> Despite operating purely as a post-hoc layer on top of FedAvg, LP-FT achieves the best performance across all three metrics, outperforming even the strongest process-integrated PFL methods in both local and global accuracy. These results reinforce LP-FT as a **plug-and-play baseline** that preserves global generalization while delivering effective personalization.
> In addition, we expanded **Appendix A.1 (Related Work)** to include a broader discussion comparing PFT and PFL methods, as suggested by Reviewer 6AJx.
>
> **3. Expanded Analysis of Parameter-Efficient Fine-Tuning (PEFT).**
> We revised and completed **Section D.1, *Exploration of Parameter-Efficient Fine-Tuning under Federated Feature Distortion***, which investigates whether PEFT methods can implicitly mitigate the distortion phenomenon identified in this work.
> Based on experiments with **LoRA** and **Adapter** on **DomainNet with ViT**, Table 7 shows that while PEFT methods achieve strong local accuracy, they suffer substantial drops in global accuracy, indicating significant distortion of shared federated features.
>
> **Note:** To improve cohesion, we added a complementary discussion in the Empirical Results section of the **main manuscript** (last paragraph of Sec. 3.4) summarizing the PEFT and PFL comparisons and clarifying their role relative to the main focus of the paper.
>
> **4. Clarification of Theoretical Assumptions and Positioning.**
> We expanded the theoretical discussion to better clarify the scope and intent of our analysis. In particular, we added explicit discussion—motivated by the rebuttal—on the use of **simplified two-layer models** to obtain tractable insights under heterogeneity (see the paragraph following **Assumption 4.2**). We emphasize that our objective is not to provide architecture-agnostic guarantees, but rather to identify the **core mechanisms driving federated feature distortion** within this setting.
> We further clarified the role of the **isotropy assumption** for analyzing concept shift (see the paragraphs preceding **Lemma 4.3**) and the rationale behind the shared feature extractor initialization.
>
> **5. Local Versions of Theorems 4.4 and 4.5.**
> We added local-performance counterparts of **Theorems 4.4 and 4.5**, showing that analogous results hold at the client level. These results are formalized in **Remark 4.7** and **Corollaries E.1 and E.2**.
>
> **6. Dedicated Discussion of Label Shift.**
> We added a dedicated discussion clarifying the role of **label shift** in the context of federated feature distortion. We explain why label shift does not directly instantiate the feature-distortion mechanism studied in our theory and therefore is not analyzed theoretically. Nevertheless, label-shift scenarios are included in our empirical evaluation and are further discussed in the **Limitations** section.
>
> **7. Precise Definitions of Evaluation Metrics.**
> To improve clarity of the experimental setup, we added precise definitions of local and global accuracy in Section 3.3 (page 5), complementing the prior verbal explanation.
>
> **8. Proof Clarifications.**
> We added several clarifying sentences to the final part of the proof of **Theorem 4.5**, as requested by Reviewer WNJW, to improve readability and understanding.
>
> Finally, we thank the reviewers again for their thorough feedback, which has directly strengthened the paper. We would be happy to address any remaining questions or concerns.

---

### Decision · Action_Editor_aqKj · 2026-01-12

**Recommendation:** Reject

**Audience:**

Yes

**Audience Explanation:**

The settings of global and local generalization are interesting. This aspect has often been overlooked in certain contexts.

**Claims And Evidence:**

No

**Claims Explanation:**

Four reviewers have posted their decisions, and only one provided positive feedback. It seems that the definitions of local generalization performance and global generalization performance are inadequate, making it harder to understand the necessity of measuring global performance. The evidence base remains too limited to support the implied robustness and general baseline claims. Moreover, the baselines may be outdated, and their theoretical contributions are largely confined to two-layer MLPs. Therefore, the paper may require significant revision before possible publication.

**Resubmission Of Major Revision:**

The authors may consider submitting a major revision at a later time.